# µPhos: a scalable and sensitive platform for high-dimensional phosphoproteomics

Denys Oliinyk[1,2], Andreas Will [ID][1,2], Felix R Schneidmadel[1,2], Maximilian Böhme [ID][2,3], Jenny Rinke [ID][2,3], Andreas Hochhaus[2,3], Thomas Ernst [ID][2,3], Nina Hahn [ID][4,5], Christian Geis [ID][4,5], Markus Lubeck [ID][6], Oliver Raether[6], Sean J Humphrey [ID][7✉] & Florian Meier [ID][1,2✉]

## Abstract

Mass spectrometry has revolutionized cell signaling research by vastly simplifying the analysis of many thousands of phosphorylation sites in the human proteome. Defining the cellular response to perturbations is crucial for further illuminating the functionality of the phosphoproteome. Here we describe µPhos ('microPhos'), an accessible phosphoproteomics platform that permits phosphopeptide enrichment from 96-well cell culture and small tissue amounts in <8 h total processing time. By greatly minimizing transfer steps and liquid volumes, we demonstrate increased sensitivity, >90% selectivity, and excellent quantitative reproducibility. Employing highly sensitive trapped ion mobility mass spectrometry, we quantify ~17,000 Class I phosphosites in a human cancer cell line using 20 µg starting material, and confidently localize ~6200 phosphosites from 1 µg. This depth covers key signaling pathways, rendering sample-limited applications and perturbation experiments with hundreds of samples viable. We employ µPhos to study drug- and time-dependent response signatures in a leukemia cell line, and by quantifying 30,000 Class I phosphosites in the mouse brain we reveal distinct spatial kinase activities in subregions of the hippocampal formation.

**Keywords** Drug Response; Mass Spectrometry; Phosphoproteomics; Sample Preparation; Signaling
**Subject Categories** Methods & Resources; Proteomics

## Introduction

Protein phosphorylation is a widespread post-translational modification (PTM) that reversibly regulates cellular processes through a complex intracellular network of kinases and phosphatases (Olsen et al, 2006). At least three quarters of proteins in the cell are phosphorylated and dysregulated phosphorylation is associated with numerous complex diseases including cancer (Sharma et al, 2014; Hanahan and Weinberg, 2011; Jayavelu et al, 2022; Ochoa et al, 2020; Franciosa et al, 2021; Viéitez et al, 2022). Advances in mass spectrometry (MS) instrumentation, sample preparation, and bioinformatics have enabled the study of protein phosphorylation dynamics on a system-wide scale (Riley and Coon, 2016; Kitata et al, 2021). As witnessed in proteomics, where rapid high-coverage proteomes can now be obtained for various organisms including mammalian cells (Mann et al, 2013; Aebersold and Mann, 2016), advances in MS technologies are shifting the focus of phosphoproteomics from a 'discovery' mode, towards functionally characterizing the myriad of phosphorylated sites and their kinase-substrate relationships (Needham et al, 2022; Zecha et al, 2023; Leutert et al, 2023). This is a particularly ambitious task considering that <5% of the more than 100,000 phosphosites currently reported have experimentally-defined cognate kinases, and even fewer are functionally assigned (Needham et al, 2019). There is therefore a growing need to further increase the throughput, sensitivity, and robustness of MS-based phosphoproteomics workflows to facilitate higher-dimensional experimental designs and to model signaling networks in greater detail (Leutert et al, 2023; Zecha et al, 2023; Needham et al, 2019).

A pivotal development in proteomics has been the adoption of data-independent acquisition (DIA) methods, which acquire peptide fragment ions across chromatographic elution profiles by cycling through predefined isolation windows encompassing the full precursor mass range (Ludwig et al, 2018; Gillet et al, 2012). While generally producing highly consistent datasets, co-isolation of multiple precursors within relatively wide isolation windows also presents data processing challenges (Ting et al, 2015; Lou et al, 2023; Fröhlich et al, 2022). Applications of DIA to phosphoproteomics have demonstrated accurate and reproducible quantification of thousands of phosphosites in time (Salovska et al, 2023; Tanzer et al, 2021) and space (Martinez-Val et al, 2021) for hundreds of samples per study. We have recently shown that combining trapped ion mobility spectrometry (TIMS) and parallel accumulation–serial fragmentation (PASEF) with DIA (dia-PASEF) enables rapid phosphoproteome measurements without sacrificing depth or sensitivity (Oliinyk and Meier, 2022; Skowronek et al, 2022; Meier et al, 2021).

As data acquisition speed and robustness improve, scaling sample processing workflows to accommodate more samples becomes a

[1]Functional Proteomics, Jena University Hospital, 07747 Jena, Germany. [2]Comprehensive Cancer Center Central Germany, 07747 Jena, Germany. [3]Klinik für Innere Medizin II, Jena University Hospital, 07747 Jena, Germany. [4]Section of Translational Neuroimmunology, Department of Neurology, Jena University Hospital, 07747 Jena, Germany. [5]Center for Sepsis Control and Care, Jena University Hospital, 07747 Jena, Germany. [6]Bruker Daltonics GmbH & Co. KG, 28359 Bremen, Germany. [7]Murdoch Children's Research Institute, Royal Children's Hospital, Parkville 3052 Victoria, Australia. ✉E-mail: sean.humphrey@mcri.edu.au; florian.meier@med.uni-jena.de

bottleneck. Selective enrichment of phosphorylated peptides from complex biological samples is well-established in proteomics for various affinity chemistries, including immobilized metal cations and metal oxide particles (Fílla and Honys, 2012; Low et al, 2021). Recent advances focused on parallelizing and streamlining the enrichment of phosphorylated peptides (Post et al, 2017; Bekker-Jensen et al, 2020; Tape et al, 2014; Riley and Coon, 2016; Leutert et al, 2019; Humphrey et al, 2018). However, typical experiments still start with millions of cells per condition to achieve sufficient phosphoproteome depth to cover key regulatory sites, which entails high cost, complexity, and significant hands-on processing time. In other words, while enrichment and data acquisition now scale to hundreds of samples, biological studies remain largely limited to classic low-throughput formats due to insufficient sensitivity. The widely used EasyPhos workflow, for example, was originally designed for input amounts of >1 mg protein (Humphrey et al, 2015a). Eliminating protein-precipitation steps and optimizing buffers for smaller input amounts facilitates phosphopeptide enrichment from hundreds of μg (Humphrey et al, 2018). However, this is still equivalent to about one million cells per sample (Rosenberger et al, 2023). Using smaller sample amounts presents challenges due to lossy sample transfer steps and high volume-to-sample ratios, often requiring compromises in accessibility, robustness, and scalability (Koenig et al, 2022; Masuda et al, 2011; Tsai et al, 2023; Yang et al, 2023; Bortel et al, 2024). Here we introduce μPhos ('microPhos'), a cost- and resource-efficient platform for lossless processing of up to 96 samples in plate format, starting from as few as tens of thousands of cells. We demonstrate the potential of this workflow for systems biology by rapidly quantifying drug- and time-dependent response signatures in a cancer cell line, and by delineating the phosphoproteome of small anatomical regions in the mouse brain.

# Results

## Design of a scalable and sensitive phosphoproteomics workflow

Envisioning high-dimensional perturbation studies as a prime application for μPhos, we first sought to determine the sensitivity required to

obtain comprehensive phosphoproteomes from multi-well cell culture experiments. A single well of a standard 6-well plate contains around 800,000 cells, corresponding to ~200 μg of protein (the typical input material used for EasyPhos), while 40,000 cells from one well of a 96-well plate yields around 10 μg protein (Fig. EV1A). Therefore, we reasoned that it would be necessary to increase the efficiency of the entire phosphoproteomics pipeline from cells to MS by an order of magnitude compared to established phosphoproteomics workflows. In a manner analogous to concepts explored in single-cell proteomics (Kelly, 2020), we designed μPhos (i) to reduce all operation volumes from ~1 mL to ~100 μL, (ii) to process 96 samples in parallel while minimizing transfer steps and hands-on time, (iii) to be compatible with low-input up-front workflows, and (iv) to avoid reliance on specialized equipment or reagents, maximizing accessibility of the workflow (Fig. 1).

As a further bottleneck towards handling many samples, we examined the timing of a typical phosphoproteomics experiment (Fig. EV1B). In our hands, processing a full plate of 96 samples according to the EasyPhos protocol excluding digestion time required on average around 500 min (5.2 min/sample). A substantial fraction of this time was consumed by upstream sample preparation steps, including cell harvesting, lysis, and the routinely performed high-energy sonication step for DNA shearing. This step, unless using specialized instrumentation, cannot be readily parallelized for >12 samples, presenting a barrier for up-scaling. However, when working with lower input materials in smaller volumes, we found water-bath sonication could replace this step without impacting sample quality (see below). We also found it possible to omit protein concentration determination and several transfer steps. As a result, we reduced the average time per sample by ~10-fold (0.5 min/sample) with the μPhos workflow, which substantially improves the feasibility of preparing many samples in large-scale studies (Fig. EV1C). Overall, these improvements should enhance scalability and reproducibility, and resulting phosphoproteome data quality.

## Development of a reproducible one-day protocol in small volumes

To assess the effect of minimized working volumes in our setup, we started from the EasyPhos protocol (Humphrey et al, 2018) and

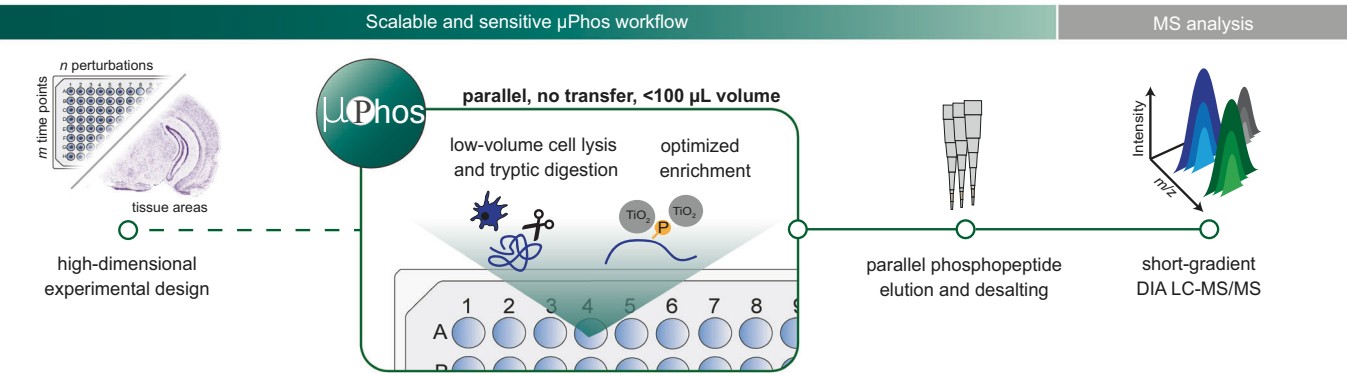

**Figure 1. Design of the μPhos platform.**

Schematic overview of the μPhos workflow for high-dimensional phosphoproteomics. The protocol streamlines cell lysis, proteolytic digestion, and phosphopeptide enrichment in low volumes, allowing it to be performed in a single reaction vessel in plate format with high sensitivity. Enriched phosphopeptides are eluted from TiO₂ beads and desalted with StageTips, and subsequently analyzed by LC-MS/MS. The image from the mouse brain section was adapted from the Allen Mouse Brain Atlas (http://atlas.brain-map.org/).

progressively decreased the total volume during phosphopeptide enrichment. Up to a ten-fold volume reduction we observed a substantial increase in the median fragment ion current, plateauing at 80 μL (Fig. EV2A). Accordingly, we observed an almost linear increase in the number of identified peptides (Fig. 2A). Investigating the peptide signal in more detail showed that the median phosphopeptide intensity increased 4-fold by decreasing the enrichment volume from 800 μL (i.e., EasyPhos) to 40 μL (Fig. EV2B). However, the relative abundance of unmodified peptides increased more strongly than that of phosphorylated peptides, resulting in a lower enrichment selectivity with decreasing enrichment volumes (Fig. EV2C). Thus, although our data supported the concept of increasing sensitivity by minimizing enrichment volumes, this came at the expense of reduced enrichment selectivity. Bearing this in mind and to ensure

compatibility with standard 96-well plates, we chose 80 μL as our working volume for subsequent experiments.

It is well-established that the binding equilibrium of phosphorylated peptides to $TiO_2$ can be influenced by various modifiers and bead binding capacity (Sugiyama et al, 2007; Larsen et al, 2005; Jensen and Larsen, 2007). Hence, we systematically screened different $TiO_2$ bead amounts as well as different concentrations and combinations of common selectivity modifiers including monopotassium phosphate ($KH_2PO_4$) and organic acids (Fig. EV2D,E). To balance selectivity with the number of identified phosphopeptides, we independently varied the final concentrations of glycolic acid and $KH_2PO_4$ from 0 to 3 M and 0 to 10 mM, respectively (Fig. 2B). When used as single agents, both compounds performed sub-optimally. However, combining at least 2 M glycolic acid and 5 mM $KH_2PO_4$ resulted in a robust plateau with a

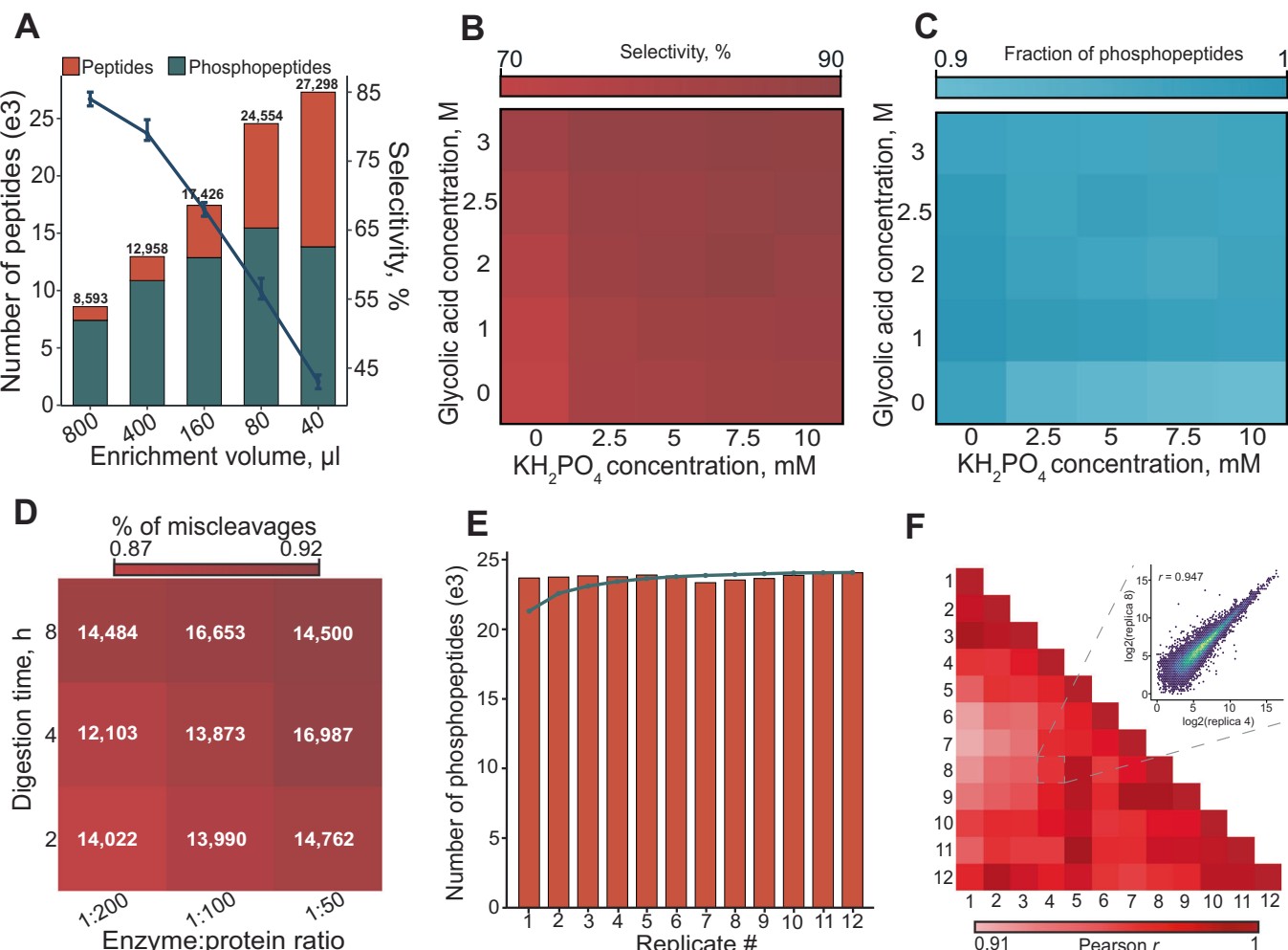

**Figure 2. Development of the μPhos platform.**

(A) Number of identified unmodified peptides and phosphopeptides for decreasing enrichment volumes starting from 20 μg HeLa cell lysates ($n = 3$). The line plot shows the selectivity of the phosphopeptide enrichment in percent. Error bars indicate the absolute variance of selectivity values. (B) Enrichment selectivity matrix using glycolic acid and $KH_2PO_4$ as selectivity agents ($n = 3$ per condition). (C) Same as (E), but showing the relative number of identified phosphopeptides. (D) Digestion efficiency and mean number of identified phosphopeptides in a matrix of incubation time and enzyme:protein ratio for starting amounts of 10 μg HeLa lysate ($n = 3$). (E) Number of identified phosphopeptides in twelve workflow replicates starting from 10 μg HeLa lysate. The blue line indicates the cumulative number. (F) Pairwise Pearson-correlation matrix for phosphopeptides quantified in twelve workflow replicates starting from 10 μg HeLa lysate and digesting for 4 h with a 1:50 protein:enzyme ratio. The inset shows a scatter plot for a pair of replicates with a correlation coefficient close to the median.

selectivity >85% and phosphopeptide identifications varying <5% (Fig. 2C). We thus selected 2.5 M glycolic acid and 5 mM $KH_2PO_4$ as our default enrichment modifiers. Importantly, testing different lysis buffers and 96-well plate types, we found the performance to be robust in terms of selectivity and reproducibility (Appendix Fig. S1), suggesting a wider flexibility with regards to lysis buffer composition and reaction containers.

The minimal hands-on time required by our protocol encouraged us to test the possibility of completing the entire workflow within one day. We therefore digested 10 μg of HeLa cell lysates for 2, 4, and 8 h, using three different enzyme-to-protein ratios ranging from 1:50 to 1:200 for both LysC and trypsin (Fig. 2D). All conditions resulted in a similar number of phosphopeptides with no clear trend along either axis, but the fraction of phosphorylated peptides with more than one mis-cleavage decreased slightly upon increased digest duration and enzyme concentration. We concluded that a digestion duration of 4 h at an enzyme-to-protein ratio of 1:50 yields an adequate digestion efficiency (<8% peptides with >1 internal lysine or arginine), while still allowing the entire protocol to be completed within one working day. The physicochemical properties of identified peptides were virtually identical even between the 2 h and 8 h digestions, and all digestion conditions yielded similar quantitative results, with median coefficients of variation around 25% in three replicates (Fig. EV2F,G).

Considering large-scale applications, we next evaluated the reproducibility of μPhos for larger sample numbers. We prepared a batch of 12 workflow replicates from 10 μg aliquots of one HeLa cell lysate with the one-day protocol. As with previous results, we observed a highly consistent performance, identifying 23,000 ± 150 phosphopeptides per replicate (Fig. 2E), resulting in an almost complete quantification matrix (97.5% reported intensity values). In total, 12,477 phosphosites were localized with a probability score >0.75. Comparing phosphopeptide intensities pairwise across replicates, we found a median Pearson correlation coefficient of 0.93 (Fig. 2F) and a median coefficient of variation of 19%.

Finally, we asked whether the optimized digestion and enrichment parameters are robust to more complex tissue phosphoproteomes (Fig. EV2H–J). Analyzing a set of murine tissue lysates (brain, heart, eye), we found <35% peptides with internal lysines or arginines and enrichment selectivities >70% for 20 μg protein input amounts. Excellent reproducibility was also evident from median coefficients of variation between 10% and 20%.

Taken together, we conclude that μPhos enables phosphopeptide enrichment from limited starting materials without compromising selectivity or robustness, while allowing all steps from cell culture to phosphopeptide enrichment to be performed in a single-plate format.

## Application to limited sample amounts

Our results indicate that μPhos is compatible with significantly lower input amounts than those typically used in phosphoproteomics studies. To determine the platform's ultimate sensitivity limits, we performed a dilution series experiment ranging from 20 to 1 μg aliquots of a HeLa cell lysate. For the analysis, we turned to the timsTOF Ultra mass spectrometer. The Ultra's brighter ion source led to a consistently increased ion current and phosphosite

intensity for 20 to 1 μg starting material compared with the timsTOF HT system (Fig. 3A). On this platform, we identified >30,000 phosphosites from 20 μg out of which ~17,000 were confidently localized (localization score >0.75) (Fig. 3B). With half the input protein we still identified >25,000 sites (12,000 Class 1) and with a quarter only about 5000 less. Starting from 1 μg protein (equivalent to the protein mass of ~4000 HeLa cells), we identified >10,000 sites of which ~6200 were confidently localized on 1795 proteins.

To test whether losses during sample processing or MS sensitivity principally limits phosphoproteome coverage at very low input levels, we repeated the experiment, but this time we prepared the dilution series after, rather than before the phosphopeptide enrichment (Fig. 3B, lower panel). Bypassing the enrichment resulted in only slightly higher numbers of identified phosphopeptides and there was no evidence of bias in the μPhos enrichment efficiency as a function of the starting amount. This is further supported by the number of identified phosphosites as a function of the Spectronaut localization probability for different input amounts, which we examined as a proxy for spectrum quality (Fig. 3C). Although the absolute numbers are different, the curves were similarly shaped and the relative proportion of confidently localized sites with a probability score cut-off >0.75 or even 0.99 remained constant. To determine the extent to which these results were dependent on the software used, we re-analyzed the dilution series using DIA-NN (Demichev et al, 2020). This yielded similar results for input amounts of 1 μg with ~4000 and ~6200 Class 1 phosphosites identified by DIA-NN and Spectronaut, respectively, and ~30% difference in the number of localized phosphosites for 20 μg input (Appendix Fig. S2a). Comparing the base peptide sequences regardless of their modification, we found an overlap of 8974 peptides (Appendix Fig. S2b).

To demonstrate the biological information that could be obtained from low input amounts, we next plotted the Class I phosphosites detected from 20 μg HeLa cell lysate ranked by their abundance (Fig. 3D). The sites spanned five orders of magnitude in relative abundance and included ~50% of all sites for which commercial antibodies are annotated in PhosphoSitePlus (Hornbeck et al, 2015). A recent bioinformatic analysis of the human phosphoproteome assigned ~11,000 sites with a high functional score (Ochoa et al, 2020). Our data covered about one-third of these, which implies a high functional relevance, especially considering that we analyzed a single, unperturbed cell line. To illustrate this further, we focused on the well-studied epidermal growth factor receptor (EGFR) signaling pathway. Figure 3E shows curated phosphosites in PhosphoSitePlus of core members of this signaling pathway, as well as upstream or downstream kinases. Almost all of these sites were detected across the complete dilution series with high confidence. From this we concluded that the combination of μPhos with highly sensitive TIMS-MS reliably quantifies biologically relevant phosphosites from ≥4000 cells (~1 μg)—less than what is typically used for immunoblotting of a single phosphosite.

## Subregion-resolved phosphoproteomics of the mouse brain

We reasoned that the scalability and sensitivity of μPhos should be advantageous beyond cell line proteomes. To illustrate this, we set

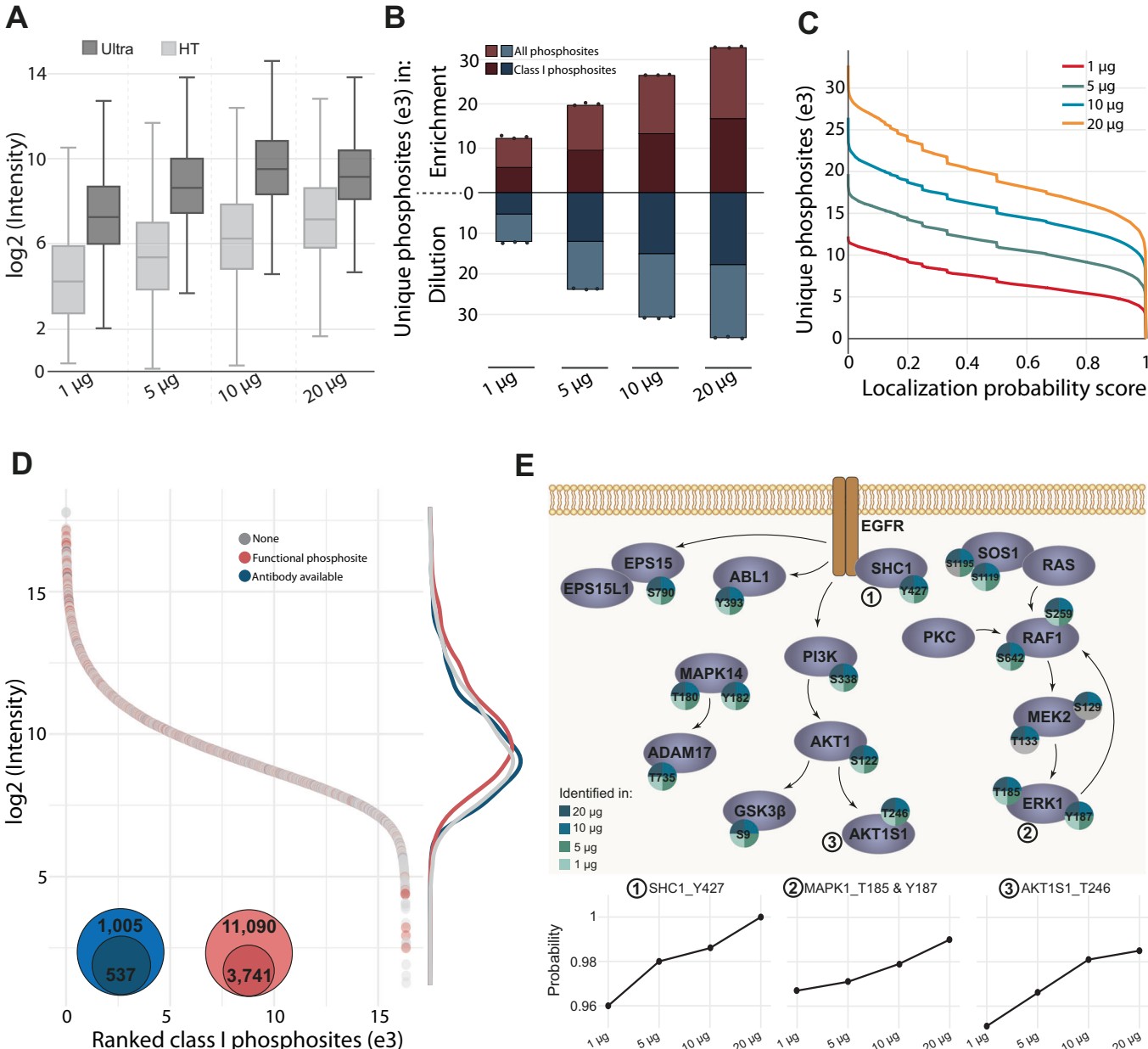

**Figure 3. Sensitive enrichment of phosphopeptides from low-μg starting amounts.**

(**A**) The box depicts the interquartile range with the central band representing the median value of the dataset. The whiskers represent the furthest datapoint within 1.5 times the interquartile range. (**B**) Number of identified phosphosites and Class 1 phosphosites (localization score >0.75, darker color) with the timsTOF Ultra as a function of input amount of HeLa cell lysate. The upper panel shows results for phosphopeptide enrichment performed with increasing starting amounts and the lower panel show results for a dilution series of phosphopeptides enriched from 20 μg starting amount. Individual data points ($n = 3$) are overlaid. (**C**) Number of phosphosites as a function of the Spectronaut site localization probability score for different protein input amounts before enrichment. (**D**) Ranked abundance of Class 1 phosphosites detected from 20 μg of a HeLa cell lysate. Venn diagrams and density plots indicate the overlap with commercially available antibodies referenced in PhosphoSitePlus (Hornbeck et al, 2015) (red) and sites in the human phosphoproteome with a predicted high functional score (Ochoa et al, 2020) (blue). (**E**) Scheme of selected phosphorylation sites associated with the Gene Ontology biological process term 'EGFR signalling pathway' and quantified in μPhos experiments starting from 1, 5, 10, and 20 μg lysate. Phosphosite localization probability scores are shown for selected phosphosites as a function of input amount.

out to map the phosphoproteome of small mouse brain regions. The fine-grained cellular and spatial organization of the brain underlies its diverse physiological functions, which is also reflected at different molecular levels (Sjöstedt et al, 2020). However, limited sample amounts have so far constrained large-scale spatial

phosphoproteomics studies to major anatomical regions, leaving finer details elusive (Liu et al, 2018). Here, we dissected seven anatomically distinct areas from six adult mice, including visual cortex (VC), entorhinal cortex (EC), corpus callosum (CC), and cerebellum (CB), as well as three subregions of the hippocampal

formation (dentate gyrus (DG), Cornu ammonis 1 (CA1) and Cornu ammonis 3 (CA3)) (Fig. 4A). Tissue volumes of the latter ranged between 0.2–0.6 mm³ and by sub-sampling these regions we were able to extract 10 to 40 μg of protein material. Despite this low input, we identified ~70,000 unique phosphosites in total and more than 30,000 phosphosites in the hippocampus alone, mapping to 6043 and 3891 phosphoproteins, respectively (Fig. 4B). We found the one-day μPhos protocol readily applicable to such small tissue amounts, resulting in an excellent reproducibility across animals with median CVs of ~30% (Fig. EV3A,B).

To obtain an overview of the brain's phosphoproteome architecture, we performed a principal component analysis (Fig. 4C), which recapitulated the previously reported separation of the cerebrum regions (cortex, hippocampus) from the cerebellum (Liu et al, 2018), while the corpus callosum formed a distinct cluster. Samples from the entorhinal cortex, the interface of the trisynaptic circuit, clustered closer to the hippocampal formation than the visual cortex and, interestingly, the CA1 and CA3 subregions of the hippocampal formation were separated from

the dentate gyrus. The latter likely reflects differences in their cellular composition as both CA subregions are rich in glutamergic pyramidal neurons, while the DG is densely populated with granule cells. These findings were further corroborated by a pairwise Pearson correlation analysis, which revealed a decreasing correlation for developmentally distant brain regions (Fig. 4D). To investigate phosphoproteome heterogeneity in the hippocampal formation in more detail, we analyzed the DG, CA1, and CA3 samples separately (Fig. EV3C,D). Integrating our data with experimentally derived substrate sequence specificities of serine/ threonine kinases (Johnson et al, 2023) indicated similar basal kinase activities in CA1/CA3, and a distinct kinase activity profile for the DG (Fig. 4E). Kinases with inferred elevated activity ($log_2$ fold-change: 0.8) included two subunits of the calcium/calmodulin-dependent protein kinase type II (Camk2), which is strongly associated with neurogenesis and synaptic plasticity (Yasuda et al, 2022) and thereby in line with the physiological function of the DG (Hochgerner et al, 2018). The DG phosphoproteome was further enriched for recognition motifs of G protein-coupled receptor

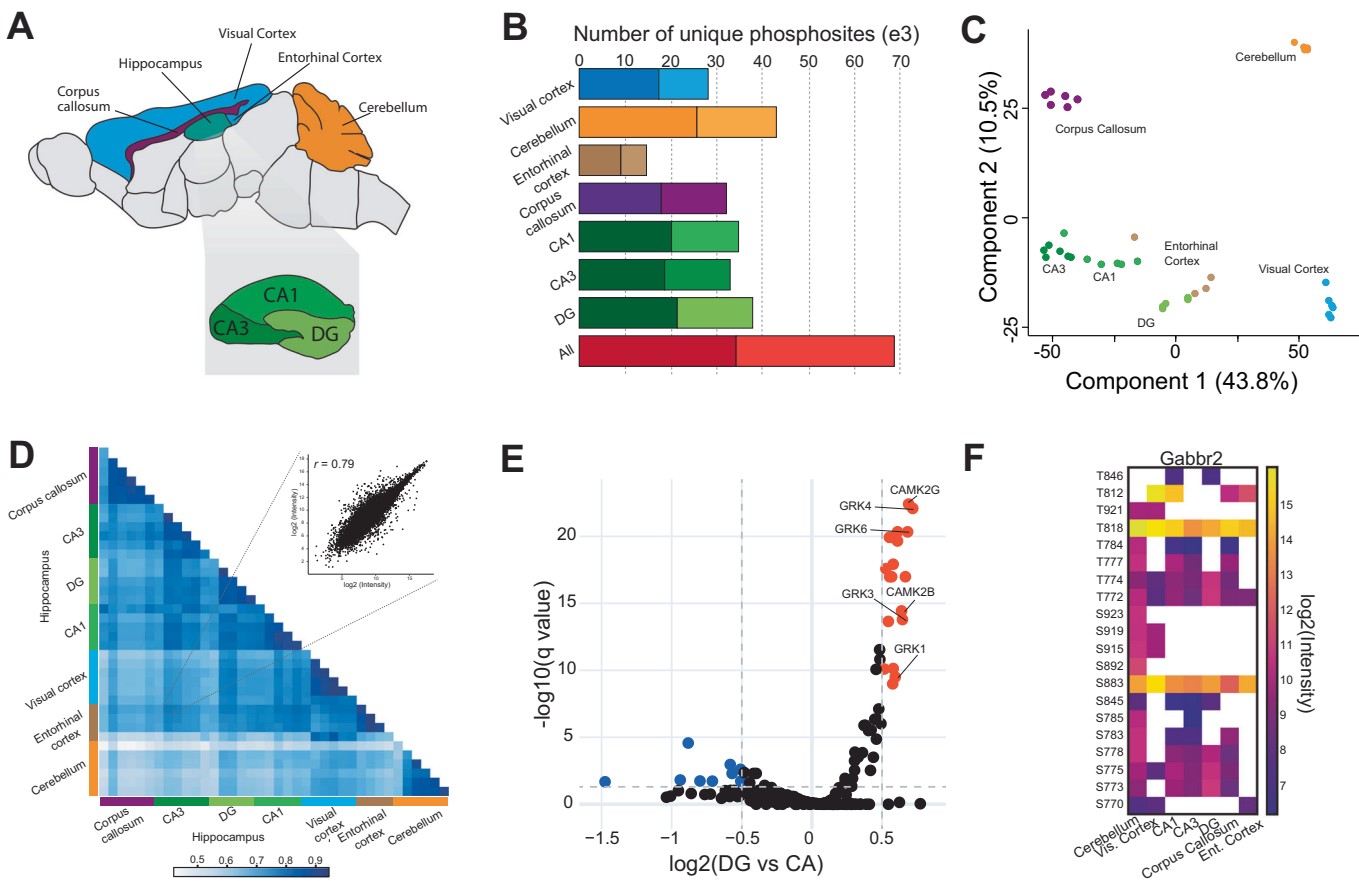

**Figure 4. Phosphoproteome analysis of murine brain subregions.**

(A) Schematic drawing of the analyzed anatomical regions. The inset zooms into the hippocampal formation. (B) Number of identified phosphosites and Class I phosphosites by anatomical region in $n = 5$–6 mice (localization score >0.75, darker color). (C) Principal component analysis, showing the first and second component. (D) Pairwise Pearson correlation analysis of all samples. The inset shows a scatter plot for entorhinal cortex vs. CA3. (E) Volcano plot of inferred kinase activities based on sequence recognition motifs (Johnson et al, 2023). Colored points indicate kinases with FDR-corrected q-values < 0.05 and a minimum absolute log2 fold-change of 0.5. We compared both CA1 and CA3 regions ($n = 12$) vs. DG ($n = 6$). (F) Heatmap of phosphosite abundance on Gabbr2 across different brain regions. The mean value of $n = 5$–6 biological replicates is shown.

kinases (GRKs)—primary regulators of G protein-coupled receptor (GPCR) sensitivity. The mismatch between >800 GPCRs and only seven known GRKs led to the hypothesis of phosphorylation 'barcodes' on the cytosolic GPCR domain as a mechanism of modulating downstream signaling (Tobin et al, 2008; Nobles et al, 2011; Lau et al, 2011). This motivated us to examine whether μPhos is capable of quantifying phosphorylation events on GPCRs. As an example, we inspected the Gamma-aminobutyric acid type B receptor subunit 2 (Gabbr2), which is broadly expressed in astrocytes (Perea et al, 2016). Remarkably, we detected phosphorylation on 20 out of 34 Ser/Thr sites in the protein C-terminus, which essentially covers the intracellular domain of the receptor (Fig. EV3E) and revealed a distinct phosphorylation pattern for each brain region (Fig. 4F). Collectively, these results pave the way for studies of receptor phosphorylation and downstream signaling events in vivo with unprecedented detail.

## Time-resolved phosphoproteomics of response to tyrosine kinase inhibitors

To further challenge the scalability of μPhos, we also applied it to study time-resolved drug responses in murine pro-B lymphocyte Ba/F3 BCR::ABL1 positive cells, a pre-clinical model of chronic myeloid leukemia (CML). CML is characterized by the BCR::ABL1 fusion gene, resulting in a constitutively active tyrosine kinase that is therapeutically targeted by small molecule inhibitors (García-Gutiérrez and Hernández-Boluda, 2019). We selected imatinib, dasatinib, nilotinib, and asciminib as frequently prescribed tyrosine kinase inhibitors (TKIs) in CML, which vary in their potency, selectivity as well as mechanism of inhibition (Hochhaus et al, 2007; Hochhaus and Kantarjian, 2013; Hochhaus et al, 2017; Iqbal and Iqbal, 2014; Amarante-Mendes et al, 2022). After seeding 50,000 cells per well in 96-well format for 24 h, we incubated them with $IC_{50}$ concentrations of the four TKIs and controls for 1, 3, 6, and 12 h, with four biological replicates for each condition (Fig. 5A; Appendix Fig. S3). μPhos enabled us to process the resulting 96 samples within one working day from cells to ready-to-inject peptides for proteome and phosphoproteome analysis. To complete the entire experiment within one week, we analyzed the 96 phosphoproteome samples with 60 min, and 96 proteome samples with 21 min gradients by label-free dia-PASEF MS. Reassuringly, all samples (except for 1 outlier) were of very high technical quality as evidenced by reproducible LC pressure profiles, with an average phosphopeptide enrichment selectivity of >90%. Library-free DIA analysis in Spectronaut identified 6245 protein groups, and ~41,000 phosphopeptides (31,000 ± 250 on average) corresponding to 23,666 unique phosphosites localized on 4151 phosphoproteins with a probability score >0.75 (Figs. 5B and EV4A,B). This translates into an identification rate of ~1250 phosphopeptides per minute in the active part of the gradient and, notably, >30,000 phosphopeptides were quantified in over three quarters (80 out of 96) of the samples (Fig. EV4C,D). The majority of the identified phosphopeptides were singly phosphorylated and at serine (81%) and threonine (15%) sites (Fig. EV4E,F). We found a median Pearson correlation of 0.91 between biological replicates and a similar correlation between drugs at each time point, while the correlation between time points was lower, indicating a time-dependent remodeling of the Ba/F3 phosphoproteome (Fig. 5C). Consistent with this, replicates and drugs grouped together in a

principal component analysis and hierarchical clustering, with most variability attributed to the time course. The comparison between proteome and phosphoproteome revealed a very weak correlation, which suggests that the observed changes in the phosphoproteome were not driven by protein abundance (Fig. EV4G,H; Appendix Fig. S4).

To contextualize changes observed in the phosphoproteome, we mapped differentially regulated phosphorylation sites (t-test adj. $p < 0.05$ and absolute fold change >1.5) to the three major known BCR::ABL1 signaling pathways. Using the well-characterized TKI imatinib as an example, Fig. 5D shows the involvement of signaling axes downstream of BCR::ABL1, including MAPK, PI3K, and Jak-Stat. Bypassing apoptosis, BCR::ABL1 phosphorylates the transcriptional regulator Stat5 on Y699 either directly or via Jak2. As expected, inhibition of BCR::ABL1 by imatinib (as well as all other TKIs) resulted in a rapid downregulation of Stat5 Y699, accompanied by reduced phosphorylation of adapter proteins such as Grb2, Crkl, Cbl, Gab2, and Sos. To further validate the time-dependent response profiles obtained by μPhos, we investigated key phosphosites curated from the literature using site-specific antibodies. This revealed a good agreement between label-free MS quantification and immunoblotting (Fig. 5E), confirming that the accuracy of our approach is maintained for 96-well perturbation experiments.

## Time-course analysis highlights drug-specific response signatures

Given that BCR::ABL1 is the common therapeutic target of all four drugs in our model, we asked whether μPhos could also dissect more subtle variations in drug response, such as those arising from potential off-target inhibition or signaling crosstalk. Our dataset contains longitudinal profiles for 23,666 phosphosites, providing additional information to conventional measurements of log-fold changes at a single time point and for a single drug. To prioritize phosphosites by the speed and duration of their response, we calculated the time-weighted area under the curve for each phosphosite ('pAUC') (Fig. 6A). This resulted in pAUC values ranging between 0 and 91 a.u. and, confirming the suitability of this score, we found Y699 on STAT5B consistently among the 2% most inhibited sites (pAUC >35) for all TKIs (Fig. EV5A). As an estimate of the overall drug efficacy, we plotted the cumulative pAUC for all ANOVA-significant phosphosites as a function of their rank order (Fig. 6B). Interestingly, out of the ~17,000 sites in our dataset, only ~3000 accounted for 50% of the summed pAUC for each inhibitor.

To explore drug-specific phosphoproteome variations, we next defined 'core' phosphosites as those with consistent response patterns between drugs (intra-drug CV ≤ 0.2) in contrast to 'variable' phosphosites (intra-drug CV ≥ 0.75) (Fig. 6C). The core phosphosites included substrates of ABL1 (Stat5b Y699, Dok1 Y369, Cbl Y698, Crkl Y207) as well as sites phosphorylated by Cdk1, Cdk2, Gsk3b, and Akt1 (Fig. 6D). Conversely, we observed several substrates of Mapk3, Prkc and Src among the variable phosphosites. Gene Ontology (GO) analysis linked the core phosphoproteome to RNA splicing, apoptosis, protein maintenance, cell growth and differentiation, and enzyme activity, suggesting that these biological processes are commonly affected by BCR::ABL1 inhibitors (Fig. 6E). Interestingly, 'chromatin organisation', 'DNA repair' and related terms were enriched in

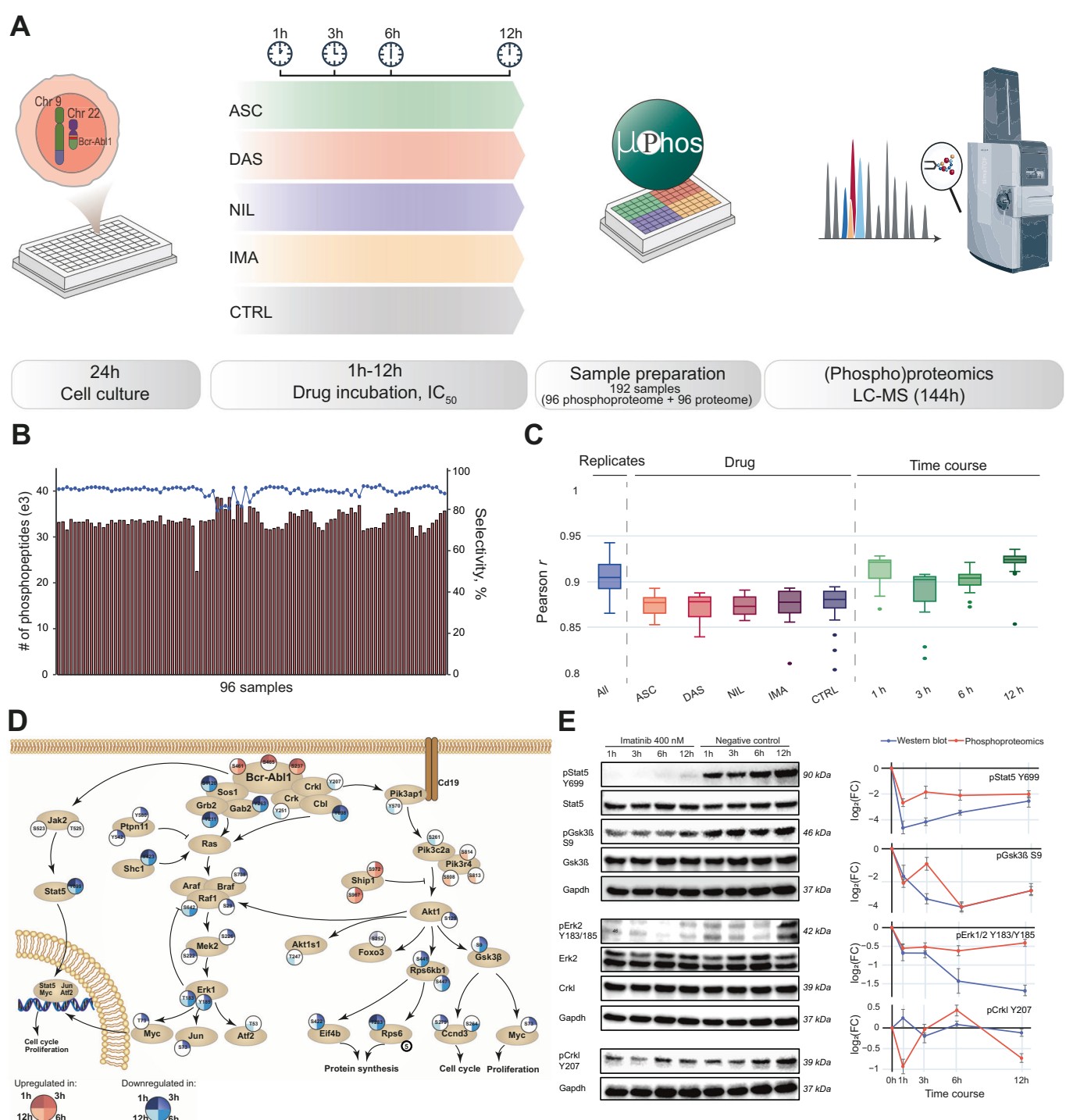

**Figure 5. Application of μPhos to study response to tyrosine kinase inhibitors.**

(A) Experimental workflow and timings. About 50,000 Ba/F3 cells expressing *BCR::ABL1* were seeded in 96-well plates and incubated with $IC_{50}$ concentrations of either imatinib (IMA), dasatinib (DAS), nilotinib (NIL), asciminib (ASC) or controls for a time course from 1 to 12 h ($n = 4$ biological replicates). Samples were processed with μPhos and subsequently analyzed with dia-PASEF single runs. (B) Number of identified phosphopeptides across 96 samples. The blue line indicates the enrichment selectivity. (C) Pairwise Pearson correlation between biological replicates ($n = 4$), per drug across time points ($n = 24$), and between drugs at each time point ($n = 16$). The box depicts the interquartile range with the central band representing the median value of the dataset. The whiskers represent the furthest datapoint within 1.5 times the interquartile range. (D) Selected pathways of *BCR::ABL1* signaling that are altered by imatinib. Colors indicate the direction of the median fold-change of shown phosphorylation sites after 1 h, 3 h, 6 h, or 12 h incubation relative to controls. (E) Validation of selected phosphorylation sites in downstream *BCR::ABL1* signaling by immunoblotting. The figure shows representative blots from $n = 3$ replicates and the corresponding Gapdh loading controls. The line plots indicate the median fold-change relative to Gapdh. Whiskers indicate the absolute inter-replicate variance ($n = 4$ for MS, $n = 3$ for immunoblotting). Source data are available online for this figure.

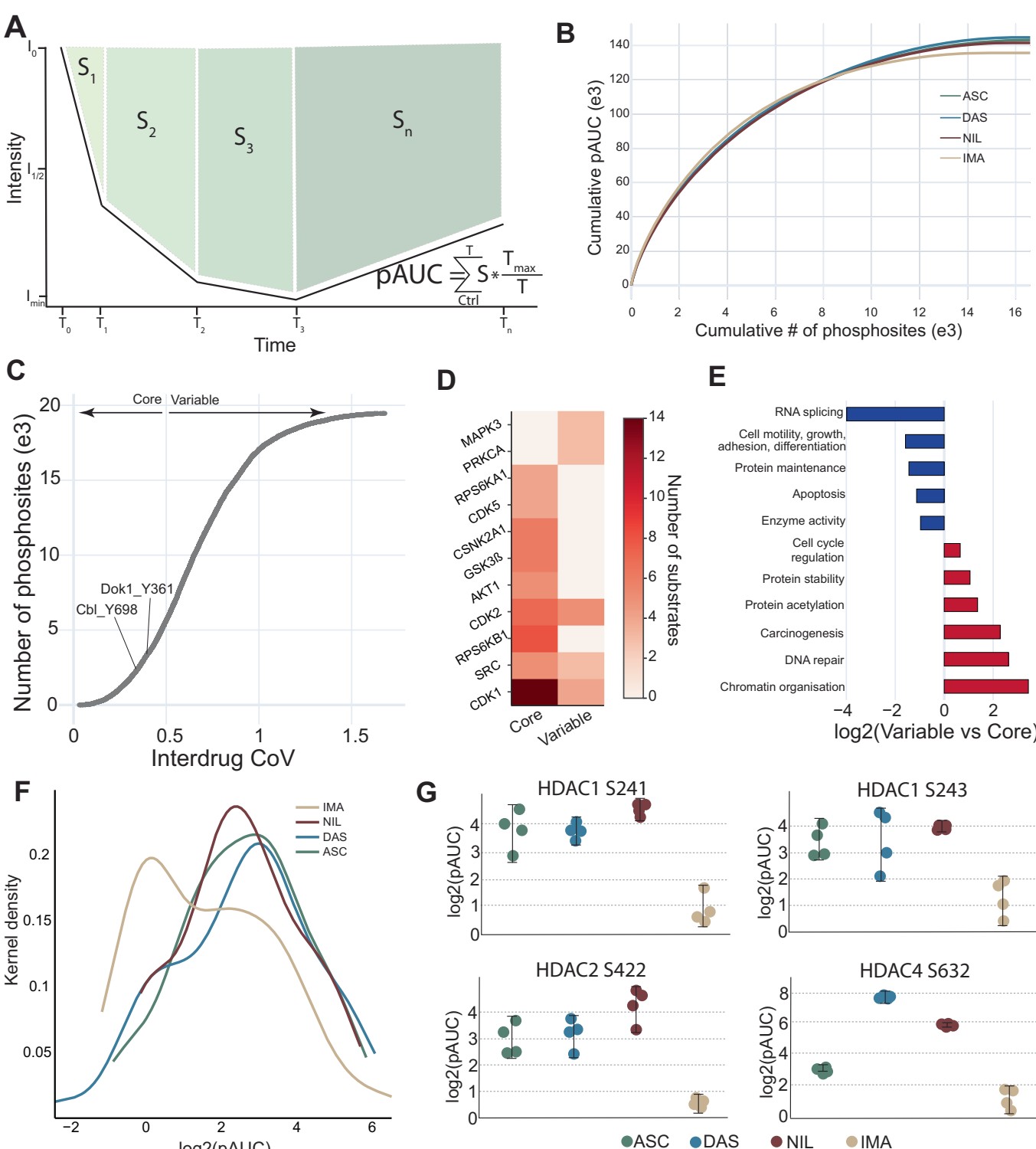

**Figure 6. Time-response signatures of tyrosine kinase inhibitors.**

(**A**) Schematic illustration of the time-weighted area under the curve (pAUC). For each phosphosite, we determined the area under the curve and multiplied it by $T_{max}/T$ to weight the speed of the drug response. (**B**) Cumulative sum of pAUC values for ANOVA-significant phosphosites as a function of their rank. (**C**) Inter-drug variability of phosphosite pAUC values. (**D**) Number of known PhosphoSitePlus substrates for selected kinases in core (CV ≤ 0.2) and variable (CV ≥ 0.75) phosphosites. (**E**) Enrichment of GO terms among core and variable phosphosites. (Benjamini-Hochberg FDR < 0.02). (**F**) Density distribution of pAUC values for phosphosites associated with the GO term "histone deacetylation". (**G**) Phosphorylation levels of HDAC sites in response to TKI treatment. Whiskers indicate the absolute variance between pAUC values of biological replicates ($n = 4$).

the variable phosphoproteome, including regulatory phosphosites on proteins of the histone deacetylation machinery. To investigate this further, we filtered for phosphosites on proteins associated with the GOBP term 'histone deacetylation' and calculated the density distribution of their corresponding pAUCs (Fig. 6F). Setting it apart from all other drugs, we observed a shift towards lower effects size (i.e., lower pAUC) for imatinib. To validate this result, we repeated our analysis for GO terms related to invariable posphosites ('MAPK signaling pathway', 'mTOR signaling pathway', 'Cell cycle' and 'Ribosome'), which resulted in indistinct density distributions for all drugs (Fig. EV5B–E). At the site level, we confirmed the dephosphorylation of regulatory sites on histone deacetylase 1 (HDAC1) and other HDACs in response to asciminib, nilotinib, and dasatinib, which was attenuated for imatinib in our experiment (Fig. 6G). This is an intriguing finding and, given the prospects for combination therapies with HDAC and tyrosine kinase inhibitors (Lernoux et al, 2020; Losson et al, 2020; He et al, 2020) demonstrates the potential to gain relevant biological insights by combining high-dimensional perturbation experiments with µPhos.

## Discussion

The transition of MS-based phosphoproteomics from mapping phosphorylation sites to their functional characterization requires increased throughput and sensitivity. To meet these demands, we developed µPhos, a robust and accessible protocol optimized for protein yields from <10,000 to 100,000 cells. Its key advantage is the miniaturization of upstream cell lysis and proteolytic digestion, enabling phosphopeptide enrichment from crude digests in <100 µL without intermediate steps in plate format. This was achieved by optimizing enrichment conditions to minimize non-specific binding of unphosphorylated peptides at high bead concentrations. Under these conditions, we also found that the duration of tryptic digestion could be reduced from overnight to just 4 h for cell culture and fresh-frozen brain tissue samples. Other sample types (e.g., muscle biopsies or formalin-fixed paraffin embedded tissue) and higher input amounts might still benefit from longer digestion or require harsher conditions for efficient lysis and protein extraction. In this case, adapting the digestion time as needed and complementing the protocol with ultra-sonication or mechanical cell lysis in plate format, e.g. similar to (Müller et al, 2020; Michaelis et al, 2023), should be straightforward, as we found that the low working volumes are compatible with different lysis buffers and 96-well plates. Together with eliminating other time-consuming steps, µPhos enables processing of 96 samples within one working day at a per-sample cost of ~$25 USD for consumables and reagents. Even ignoring per-sample costs and considering only input material requirements for phosphoproteomics and the resulting data richness, the use of immunoblotting for studying cell signaling today is now questionable.

Minimizing sample handling steps should also reduce variability within and between experiments. We found that technical reproducibility, as estimated by phosphopeptide coefficients of variation, is comparable to other state-of-the-art workflows for conventional starting amounts (Koenig et al, 2022; Martínez-val et al, 2023). Further improvements to throughput can be achieved by automation, for example using magnetic beads (Tape et al,

2014). However, such adaptations require increasingly specialized equipment and proprietary consumables. They also require re-evaluation of enrichment conditions, as different bead chemistries and physiochemical properties can strongly influence phosphopeptide enrichment performance, even under standardized conditions (Arribas Diez et al, 2021).

In addition to facilitating parallel sample processing, reducing sample volumes and minimizing exposure to surfaces generally alleviates adsorptive peptide losses, thereby increasing peptide recovery. On the same LC-MS instrument, we achieved a 4-fold increase in phosphopeptide signal compared to the EasyPhos protocol by reducing the enrichment volume from 800 µL to 80 µL. Given that phosphoproteomics experiments are often limited by signal intensity, this gain should readily transfer to other data acquisition modes and instrumentation. However, single-cell proteomics workflows routinely process samples in 5 µL or less (Matzinger et al, 2023; Thielert et al, 2022), suggesting further sensitivity gains may be achievable through continued miniaturization. Indeed, Tsai et al recently described a tip-based phosphoproteomics protocol compatible with cell lysis and digestion in <5 µL for very low input amounts (Tsai et al, 2023). Plate-based protocols are particularly attractive for applications with large sample numbers due to their scalability and robustness, however, a practical limitation is the need for sufficient volumes to collect all cells and to ensure efficient lysis and mixing during enrichment. Nonetheless, the sensitivity of our label-free analysis of low µg inputs from an unperturbed cell line using a timsTOF mass spectrometer compares favorably with other approaches tailored to minimal sample amounts (Koenig et al, 2022; Masuda et al, 2011; Tsai et al, 2023; Martinez-Val et al, 2021; Yang et al, 2023). Thus, while robust phosphoproteomics from 1 µg or less remains challenging, µPhos coupled with highly sensitive mass spectrometers shifts the working range by an order of magnitude for most practical applications, and in particular for high-dimensional experimental designs.

Advances in bioinformatics such as the integration of neural networks have made the direct analysis of DIA data without the need for experimental libraries a preferred choice in proteomics studies (Bekker-Jensen et al, 2020; Demichev et al, 2020; Lou et al, 2023; Bruderer et al, 2015; Lou et al, 2021). Given the combinatorial challenges in phosphoproteomics, library-based approaches may provide superior coverage for small sample numbers, but this advantage becomes less pronounced as sample numbers increase. We demonstrate this by both the excellent coverage of the mouse brain phosphoproteome from minimal material as well as by comprehensive coverage of *BCR::ABL1* signaling in leukemic cells. The latter allowed us to generate time-resolved drug-specific phosphoproteome signatures and characterize the mode of action of four kinase inhibitors with a rapid turn-around time of around one week. At present, label-free quantification appears most suitable for such multi-dimensional phosphoproteomics studies, as it can simply scale to compare as many samples as required (Humphrey et al, 2015b). However, recent multiplexed DIA strategies should also be readily compatible with µPhos, which may be particularly beneficial for sub-µg input amounts and for further increasing sample throughput (Koenig et al, 2022; Thielert et al, 2022; Derks et al, 2023).

We conclude that µPhos is well suited for systems-scale phosphoproteomics studies and anticipate a wide range of

applications providing new insights into cellular signaling networks, including their responses to internal or external perturbations, or as a function of their cellular and spatial context. The ability to prepare 96 phosphoproteome samples within one working day, combined with high reproducibility and coverage from minimal starting material will likely contribute to the acceleration of phosphoproteomics research, opening new avenues for studying signal transduction.

# Methods

### Reagents and tools table

| Reagent/resource | Reference or source | Identifier or catalog number |
|---|---|---|
| **Chemicals, enzymes, and other reagents** | | |
| Sodium deoxycholate | Sigma-Aldrich/Merck | D6750 |
| 2,2,2-Trifluoroethanol | Sigma-Aldrich/Merck | T63002 |
| Acetonitrile | Sigma-Aldrich/Merck | 1.00029 |
| Tris(hydroxymethyl)-aminomethan (Tris) | Sigma-Aldrich/Merck | GE17-1321-01 |
| Tris(2-carboxyethyl) Phosphine (TCEP) | Sigma-Aldrich/Merck | C4706 |
| Chloracetamide (CAA) | Sigma-Aldrich | C0267 |
| Trypsin | Sigma-Aldrich | T6567 |
| Lys-C | FUJIFILM Wako | 125-05061 |
| Isopropanol | Sigma-Aldrich/Merck | 1.02781 |
| Glycolic acid | Sigma-Aldrich/Merck | 124737 |
| Potassium monophosphate | Sigma-Aldrich/Merck | 1.04875 |
| Trifluoroacetic acid | Sigma-Aldrich | T6508 |
| Ammonium hydroxide | Sigma-Aldrich/Merck | 338818 |
| CDS-Empore SDB-RPS extraction discs | Sigma-Aldrich | 66886-U |
| CDS-Empore C8 extraction discs | Sigma-Aldrich | 66882-U |
| 500 mg Bulk Material Titansphere Phos-TiO 10 µm | MZ-ANALYSENTECHNIK GMBH | 5010-21315 |
| **Software** | | |
| RStudio (2022.07.2) and R (4.2.2) | Posit PBC | |
| Jupyter Notebook | https://jupyter.org/ | |
| Spectronaut v.17 | BiognoSYS AG | |
| DIA-NN v1.8.1 | https://github.com/vdemichev/DiaNN | |
| **Other** | | |
| 96-well deep well plate | Eppendorf | 0030502302 |
| 96-well deep well plate sealing mat | Thermo Fisher Scientific | 9503230 |
| 96-well F-bottom cell culture plate | Sigma-Aldrich/Merck | M0812 |
| 96-well U-bottom cell culture plate | Sigma-Aldrich/Merck | M9436 |
| 96-well V-bottom cell culture plate | Sigma-Aldrich/Merck | M9686 |

| Reagent/resource | Reference or source | Identifier or catalog number |
|---|---|---|
| 96-well cell culture plate sealing mat | Thermo Fisher Scientific | 60180-M179 |
| ThermoMixer | Eppendorf | 460-0223 |
| NanoDrop One/OneC Microvolume UVVis Spectrophotometer | Thermo Fisher Scientific | |
| timsTOF HT/Ultra | Bruker Daltonik GmbH | |
| Column oven | Sonation lab solutions | #PRSO-V2 |
| Bruker PepSep column | Bruker Daltonik GmbH | 1893626 |

## Methods and protocols

### Human cell culture

Human cancer cells (epithelial cervix carcinoma, HeLa, RRID: CVCL_0030) were cultured in RPMI 1640 medium supplemented with 10% fetal bovine serum, 2% penicillin/streptomycin and 5 µL/mL plasmocin in an incubator with 5% $CO_2$ at 37 °C. Cells were harvested at ~80% confluency with 0.25% trypsin/EDTA and collected in 15 mL falcon tubes. Subsequently, cells were washed twice with tris-buffered saline, pelleted by centrifugation for 2 min at $1000 \times g$ and stored at −80 °C until further use. The cell line was not authenticated in the course of this study. The cell culture was routinely tested for mycoplasma contamination.

### Drug response screen

To establish a *BCR::ABL1*-positive cell line, murine Ba/F3 cells (DSMZ, Leibniz Institute, Germany; Cat. no. ACC 300, RRID: CVCL_0161) were transfected with a plasmid containing full-length *BCR::ABL1* (transcript type b3a2) using Lipofectamine 3000 Transfection Reagent (Invitrogen). The cell culture was routinely tested for mycoplasma contamination. *BCR::ABL1* positive Ba/F3 cells (a suspension cell line) were seeded in a 96-well plate in 200 µL RPMI 1640 medium supplemented with 10% fetal bovine serum in an incubator with 5% $CO_2$ at 37 °C for 24 h before incubation with 400 nM imatinib, 2.5 nM dasatinib, 30 nM nilotinib, and 4.5 nM asciminib in 4 biological replicates for 1 h, 3 h, 6 h, or 12 h. As controls we used (1) only medium and (2) 400 nM DMSO. To avoid artefacts due to overgrowth, we first established a suitable number of seeded cells per well based on a cell toxicity assay (MTS) and cell counts after 24 h of incubation (Fig. EV4A,B). We found that cultivation of ~50,000 cells per well yielded the maximal number of viable cells after 24 h, while seeding more cells resulted in cell aggregation that may prevent efficient drug uptake and induce cellular stress response (Fig. EV4C,D). We note that the optimal cell number per well is cell line specific. We therefore recommend optimizing the cell density on a per-cell line basis prior to commencing studies.

### Tissue lysis for bulk experiments

Mice were kept according to institutional and legal regulations (§11 TierSchG). C57BL/6Jrj mice (female) were kept in individually ventilated cages (IVCs) under Specific Pathogen Free (SPF) conditions with a 12 h/12 h dark/light cycle at a temperature of 20 °C and a relative humidity of 55% according to the directives of the 2010/63/EU and GV SOLAS. Aged mice between 22 and 28

months were sacrificed and dissected organs were snap-frozen with liquid nitrogen and homogenized with pestle and mortar, precooled at 4 °C. Homogenized tissues were lysed in SDC lysis buffer (4% sodium deoxycholate, 100 mM Tris-HCl pH 8.5), incubated at 95 °C, followed by high-energy sonication (Diagenode Bioruptor pico). Protein concentration was determined via a BCA assay (Thermo Fisher). Protein disulfide bonds were reduced with tris(2-carboxyethyl)phosphine and cysteine was alkylated with 2-chloroacetamide at final concentrations of 10 mM and 40 mM for 5 min at 45 °C. 1 µL of trypsin/Lys-C mix (final enzyme:protein ratio of ~1:100) were added to each well and incubated for 4 h at 37 °C at 1500 rpm.

### Dissection and lysis of mouse brain tissue

All animal experiments have been approved by the Thuringian state authorities (authorization twz08-2020). Mice (strain C57BL/6J) were housed under controlled day/night (12 h/12 h) conditions at room temperature (23 ± 1 °C, 30–60% environmental humidity). They received a standard diet and water ad libitum in the in-house breeding facility (service center for small rodents, Jena, University Hospital). Six (3 female, 3 male) 3-month-old mice were deeply anesthetized with isoflurane and transcardially perfused with 25 mL of Tris-buffered saline (TBS) for 2 min before dissection of the brain. Note that dissection was not successful in all regions of all mice (e.g., due to full or partial infusion with blood), resulting in $n = 5$ for entorhinal cortex and dentate gyrus. Dissected tissue samples were directly transferred into 17 µL of SDC lysis buffer (4% sodium deoxycholate, 100 mM Tris-HCl pH 8.5) in 96-well plate stored at 4 °C on wet ice. Tissues were further lysed by incubation at 72 °C for 10 min at 1500 rpm, followed by sonication in a water bath sonicator (Elmasonic, S 60/H) for another 10 min at room temperature. After short centrifugation, 2 µL of 10X alkylation/reduction buffer (100 mM and 400 mM of tris(2-carboxyethyl)phosphine and 2-chloracetamide), pH 8 were added to each well and incubated for 5 min at 45 °C at 1500 rpm. Finally, 1 µL of trypsin/Lys-C mix (final enzyme:protein ratio of approximately 1:100) were added to each well and incubated for 4 h at 37 °C at 1500 rpm.

### Cell lysis for bulk experiments

Washed cell pellets were lysed in SDC buffer (4% sodium deoxycholate, 100 mM Tris-HCl pH 8.5), TFE buffer (30% 2,2,2-trifluoroethanol, 100 mM tetraethylammonium bromide) or ACN buffer (20% acetonitrile, MiliQ H$_2$O) and boiled for 7 min at 95 °C before high-energy sonication to shear DNA (Diagenode Bioruptor pico). Protein concentrations were determined via a BCA assay (Thermo Fisher). Protein disulfide bonds were reduced with tris(2-carboxyethyl)phosphine and cysteines were alkylated with 2-chloroacetamide at final concentrations of 10 mM and 40 mM for 5 min at 45 °C. 1 µL of trypsin/Lys-C mix (final enzyme:protein ratio of approximately 1:100) were added to each well and incubated for at least 2 h at 37 °C at 1500 rpm. For µPhos experiments, we kept the total volume (i.e., cell lysate, alkylation/reduction buffer, and trypsin/Lys-C mix) below 20 µL.

### Cell lysis for 96-well plate experiments

After drug incubation, 96-well plates were centrifuged for 2 min at room temperature and 1000 rpm. The supernatant was then carefully aspirated without disturbing the cell pellet. Cells were washed with 200 µL TBS and centrifuged for 2 min at room temperature and 1000 rpm. After discarding the supernatant, cells

were lysed by adding 17 µL SDC lysis buffer. We sealed the 96-well plates with a silicone mat and incubated them at 72 °C for 10 min at 1500 rpm, followed by sonication in a water bath sonicator (Elmasonic, S 60/H) for another 10 min at room temperature. After short centrifugation, 2 µL of 10X alkylation/reduction buffer (100 mM and 400 mM of tris(2-carboxyethyl)phosphine and 2-chloroacetamide) at pH 8 were added to each well and incubated for 5 min at 45 °C at 1500 rpm. Finally, 1 µL of trypsin/Lys-C mix (final enzyme:protein ratio of approximately 1:100) were added to each well and incubated for at least 2 h at 37 °C at 1500 rpm.

### µPhos sample processing (structured protocol format)

1. 20 µL of 100% isopropanol were added to the protein digest and mixed for 30 s at 1500 rpm.
   Note: Steps 1, 2, 5–7, 8, 11–12, and 15–18 can be readily performed with a multichannel pipette.
   Note: Thorough mixing is required at this point to avoid precipitation of lysis buffer.
2. 40 µL of µPhos enrichment buffer (10 mM KH$_2$PO$_4$, 12% TFA, 5 M glycolic acid in 50% isopropanol) were added and mixed for 30 s at 1500 rpm in a ThermoMixer.
3. TiO$_2$ beads were weighed and suspended in µPhos loading buffer (6% TFA in 80% ACN) at a concentration of 1 mg/µL. 5 µL of the bead suspension was added to the sample and mixed at 40 °C for 7 min at 1500 rpm.
   Note: Bead suspension must be thoroughly mixed immediately prior to pipetting to obtain a homogeneous bead distribution.
4. Beads were pelleted by centrifugation for 1 min at $2000 \times g$ and the supernatant was either stored for proteomics experiments or removed by vacuum aspiration.
   Note: Use the recommended aspiration points to efficiently remove supernatant and avoid bead loss (Appendix Fig. S1j).
5. 200 µL of µPhos washing buffer (5% TFA in 60% isopropanol) was added to the pelleted beads and mixed for 1 min at 2500 rpm at room temperature.
6. Beads were pelleted by centrifugation for 1 min at $2000 \times g$ and the supernatant was removed by vacuum aspiration.
7. Steps 5 and 6 were repeated a further three times (four washes total).
8. 75 µL of µPhos transfer buffer (0.1% TFA in 60% ACN) were added to each well and the bead suspension was transferred to C8 StageTips.
9. To minimize sample losses, step 8 was repeated.
   Note: Carefully flush the bottom of the well with the buffer to transfer all beads.
10. Transfer buffer was removed by centrifugation for 7 min at $1500 \times g$.
    Note: It is critical to remove all transfer buffer before adding the elution buffer.
11. Phosphopeptides were eluted with 30 µL of µPhos elution buffer (5% NH$_4$OH in 40% ACN) by centrifugation for 4 min at $1500 \times g$.
    Note: The elution buffer must be prepared fresh to avoid changes in pH.
12. To maximize phosphopeptide recovery, step 11 was repeated and the eluates were combined.
13. The combined eluates were vacuum centrifuged for 30 min at 45 °C until less than ~10 µL volume remained.

14. 100 μL of SDB-RPS loading buffer (1% TFA in isopropanol) was added to the reduced eluates and loaded onto SDB-RPS StageTips via centrifugation for 7 min at $1000 \times g$.

15. For washing, we first added 100 μL SDB-RPS loading buffer to each StageTip and centrifuged them for 7 min at $1000 \times g$.

16. Step 15 was repeated with 100 μL SDB-RPS washing buffer (0.2% TFA in $H_2O$).

17. Phosphopeptides were eluted from StageTips with 60 μL of SDB-RPS elution buffer (1% $NH_4OH$ in 40% ACN) and vacuum centrifuged to dryness for 40 min at 60 °C.

18. Phosphopeptides were reconstituted in 3 μL MS loading buffer (0.3% TFA, 2% ACN in $H_2O$) and stored at −80 °C until LC-MS measurement.

### Liquid chromatography-mass spectrometry analysis

Purified and desalted peptides were separated via nanoflow reversed-phase liquid chromatography (Bruker Daltonics, nanoElute) within 30 min or 60 min at a flow rate of 0.5 μL/min on a 15 cm × 75 μm column packed with 1.9 μm $C_{18}$ beads (Bruker/PepSep). Mobile phase A was water with 0.1 vol% formic acid and B was ACN with 0.1 vol% formic acid. For phosphoproteome analysis, the gradient was as follows: %B increased linearly between 2% at 0 min and 35% at 52 min, followed by a steep increase to 95% at 57 min, which was held constant for 3 min. For proteome analysis, the gradient was as follows: %B increased linearly between 2% at 0 min and 35% at 26 min, followed by a steep increase to 95% at 27 min, which was held constant for 3 min. If applicable, samples were injected in randomized order. Peptides eluting from the column were electrosprayed (CaptiveSpray) into a TIMS quadrupole time-of-flight mass spectrometer (Bruker timsTOF HT or timsTOF Ultra). We acquired all data in dia-PASEF mode (Meier et al, 2020) with an optimized isolation window scheme in the $m/z$ vs ion mobility plane for phosphopeptides (Appendix Table S1) (Skowronek et al, 2022). The ion mobility range was set from $1/K_0 = 1.43$ to $0.6$ Vs cm$^{-2}$. To maximize the ion utilization, we used equal ion accumulation time and ramp times in the dual TIMS analyzer of 100 ms each, resulting in an acquisition cycle time of ~1.34 s. The energy for collision induced dissociation was defined as a function of the ion mobility and linearly decreased from 59 eV at $1/K_0 = 1.4$ Vs cm$^{-2}$ to 20 eV at $1/K_0 = 0.6$ Vs cm$^{-2}$. For all experiments, we calibrated the TIMS elution voltages by reference $1/K_0$ values from at least two out of three ions from Agilent ESI LC/MS tuning mix ($m/z$, $1/K_0$: 622.0289, 0.9848 Vs cm$^{-2}$; 922.0097, 1.1895 Vs cm$^{-2}$ ; and 1221.9906, 1.3820 Vs cm$^{-2}$).

### Immunoblotting

Western blot analysis of signaling pathways was performed using BCR::ABL1-positive Ba/F3 cells. Cells were seeded in a 25 cm$^2$ cell flask ($1 \times 10^6$ cells/ml) in RPMI supplemented with 10% fetal bovine serum. Prior to imatinib treatment (400 nM), cells were cultivated for 24 h at 37 °C and 5% $CO_2$. Preparation of cell lysates and protein separation was performed according to standard protocols and as previously described (Schäfer et al, 2016). Primary antibodies against Stat5 mAB (#25656), phospho-Stat5 Y694 mAB (#9314), Gsk-3ß mAB (#9315), phospho-Gsk3ß S9 polyclonal AB (#9336), Crkl mAB (#38710), phospho-Crkl Y207 mAB (#34940) and Gapdh mAB (#2118) were purchased from Cell Signaling Technology (Boston, MA, USA) and incubated for 24 h at 4 °C (diluted 1:1000 in TBS-T,

pH = 7.6). After incubation with the HRP-conjugated secondary antibody (1:5000 in TBS-T, pH = 7.6), proteins were visualized by adding chemiluminescent HRP substrate (Millipore, Billerica, MA, USA) and subsequently analyzed by LI-Cor Odyssey XF (LI-COR Bioscience, Bad Homburg, Germany). For each blot, we measured Gapdh as a loading control and for internal normalization. If applicable, membranes were cut and stripped to detect multiple proteins and phosphosites in one run.

### MS data processing

dia-PASEF raw files were processed in Spectronaut v17.2 and DIA-NN v1.8.1 without experimental spectrum libraries ('directDIA+' workflow in Spectronaut). Data were searched against the UniProt human or mouse reference proteome including isoforms (accessed April and June 2022, respectively). We set the protease specificity to trypsin with a maximum number of two missed cleavages and required a minimum peptide length of 7 amino acids. The mass tolerances for precursor and fragment ions were set to 'Dynamic' for both MS1 and MS2 level. False discovery rates were controlled by a target-decoy approach to ≤1% at precursor and protein levels. We defined cysteine carbamidomethylation as a fixed modification and protein N-terminal acetylation, methionine oxidation and serine/threonine/tyrosine (STY) phosphorylation as variable modifications. We used the 'BGS Phospho PTM Workflow' in Spectronaut and activated the PTM localization mode. To report all identified phosphopeptides, we defined a localization probability score threshold of 0 and, if applicable, filtered the output on the phosphosite level as described below. Quantification values were filtered by q-value and we defined the 'Automatic' normalization mode for cross run normalization.

### Bioinformatics analysis

No blinding was done in the experiments described in this study. All data processing and analysis steps were performed in the Python programming environment (v.3.9.1). We exported tabular data in the Spectronaut 'BGS Factory Report' scheme with 'EG.PrecursorID', 'PEP.PeptidePosition', 'EG.PTMAssayProbability', 'PG.Genes' and 'PG.ProteinGroups' as additional columns and parsed the output with a custom Python implementation of the 'PeptideCollapse' plugin for Perseus (Tyanova et al, 2016). Reverse sequences, phosphosites with localization probability <0.75 and phosphosites quantified in <70% of technical/biological replicates were removed and remaining missing values were imputed by random sampling from a downshifted normal distribution as previously described (Tyanova et al, 2016). Two-way ANOVA, volcano plot analysis, unsupervised hierarchical clustering analysis and PCA were performed in Python using scripts adapted from the Clinical Knowledge Graph analytics core (Santos et al, 2022). $P$ values were corrected for multiple comparisons with the Benjamini-Hochberg (BH) method (Benjamini and Hochberg, 1995). For volcano plot analysis, we considered phosphosites significantly regulated with an adjusted $P$ value < 0.05 and an absolute fold change >1.5. Kinase activity prediction was performed using the 'Kinase Library' enrichment analysis (https://kinase-library.phosphosite.org/ea) (Johnson et al, 2023) using the default parameters in the web interface. For the analysis of the tyrosine kinase inhibitor experiment, we calculated the area under the curve for each ANOVA-significant (adj. $P$ value < 0.05) phosphosite from the normalized abundance vs. incubation time curve by integration.

To prioritize rapidly regulated sites, we weighted the area by a factor $T_{max}/T$. Gene Ontology (GO) analysis of 'core' vs. 'variable' sites was performed on the level of phosphosites (Fig. 6E) or phosphorylated proteins (Fig. 6F). Prior to the GO enrichment analysis, we mapped phosphosites from the mouse phosphoproteome to human orthologs based on the PhosphoSitePlus Phosphorylation Site Dataset (version April 2021). For the phosphosite level GO analysis, only those phosphosites with an annotated function (PhosphoSitePlus Regulatory Sites database, version April 2021) were subjected to a Fisher's exact test (Benjamini-Hochberg < 0.02) using all ANOVA-significant sites as the background.

## Data availability

The datasets produced in this study are available in the following databases: Phosphoproteomics data: PRIDE (Perez-Riverol et al, 2022) PXD043370.

The source data of this paper are collected in the following database record: biostudies:S-SCDT-10_1038-S44320-024-00050-9.

## Peer review information

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

## Acknowledgements

This work was partially supported by the Federal Ministry of Education and Research and the Thuringian Ministry for Economic Affairs, Science and a Digital Society through the Joint Federal Government-Länder Tenure-Track Programme, by the Free state of Thuringia and the European Union via the "Innovationszentrum für Thüringer Medizintechnik-Lösungen" (ThIMEDOP; #2018 IZN 002), by the Center for Interdisciplinary Clinical Research (IZKF Jena), by the German Research Foundation through the Research Training Group 'ProMoAge' (RTG 2155), by the National Health and Medical Research Foundation (NHMRC; grant #2011083), and the Stafford Fox Medical Research Foundation. We acknowledge L Reiter, T Gandhi, and O Bernhardt (BiognoSYS AG) for technical support with Spectronaut and T Müller (Bruker) for technical support and discussions. The HeLa cell line was a kind gift from the A Ori laboratory (FLI Jena). We acknowledge the support from animal breeding facilities at the UKJ and FLI. We thank S Schmidt, C Sommer, and M Haase (all UKJ) for dissection of mouse brain tissue regions and B. von Eyss (FLI Jena) for providing mouse organs for bulk analysis. We are grateful to our colleagues at the Jena University Hospital in the research group 'Functional Proteomics' for fruitful discussions and technical support, in particular D Ernafasova, P Köcher, and C Tschernjawski.

## Author contributions

**Denys Oliinyk**: Conceptualization; Formal analysis; Investigation; Visualization; Methodology; Writing—original draft; Writing—review and editing. **Andreas Will**: Formal analysis; Investigation; Writing—review and editing. **Felix R Schneidmadel**: Formal analysis; Investigation; Writing—review and editing. **Maximilian Böhme**: Formal analysis; Validation; Investigation; Visualization; Methodology; Writing—review and editing. **Jenny Rinke**: Formal analysis; Investigation; Methodology; Writing—review and editing. **Andreas Hochhaus**: Resources; Supervision; Writing—review and editing. **Thomas Ernst**: Resources; Supervision; Methodology; Writing—review and editing. **Nina Hahn**: Investigation; Methodology; Writing—review and editing. **Christian Geis**: Resources; Supervision. **Markus Lubeck**: Resources; Investigation; Methodology. **Oliver Raether**: Resources; Methodology. **Sean J Humphrey**: Conceptualization; Resources; Formal analysis; Supervision; Funding acquisition; Investigation; Visualization; Writing—original draft; Writing—review and editing. **Florian Meier**: Conceptualization; Resources; Formal analysis; Supervision; Funding acquisition; Investigation; Visualization; Methodology; Writing—original draft; Project administration; Writing—review and editing.

Source data underlying figure panels in this paper may have individual authorship assigned. Where available, figure panel/source data authorship is listed in the following database record: biostudies:S-SCDT-10_1038-S44320-024-00050-9.

## Funding

## Disclosure and competing interests statement

ML and OR are employees of Bruker Daltonics, the vendor of the mass spectrometers used in this study. All other authors declare no competing interests.

# Expanded View Figures

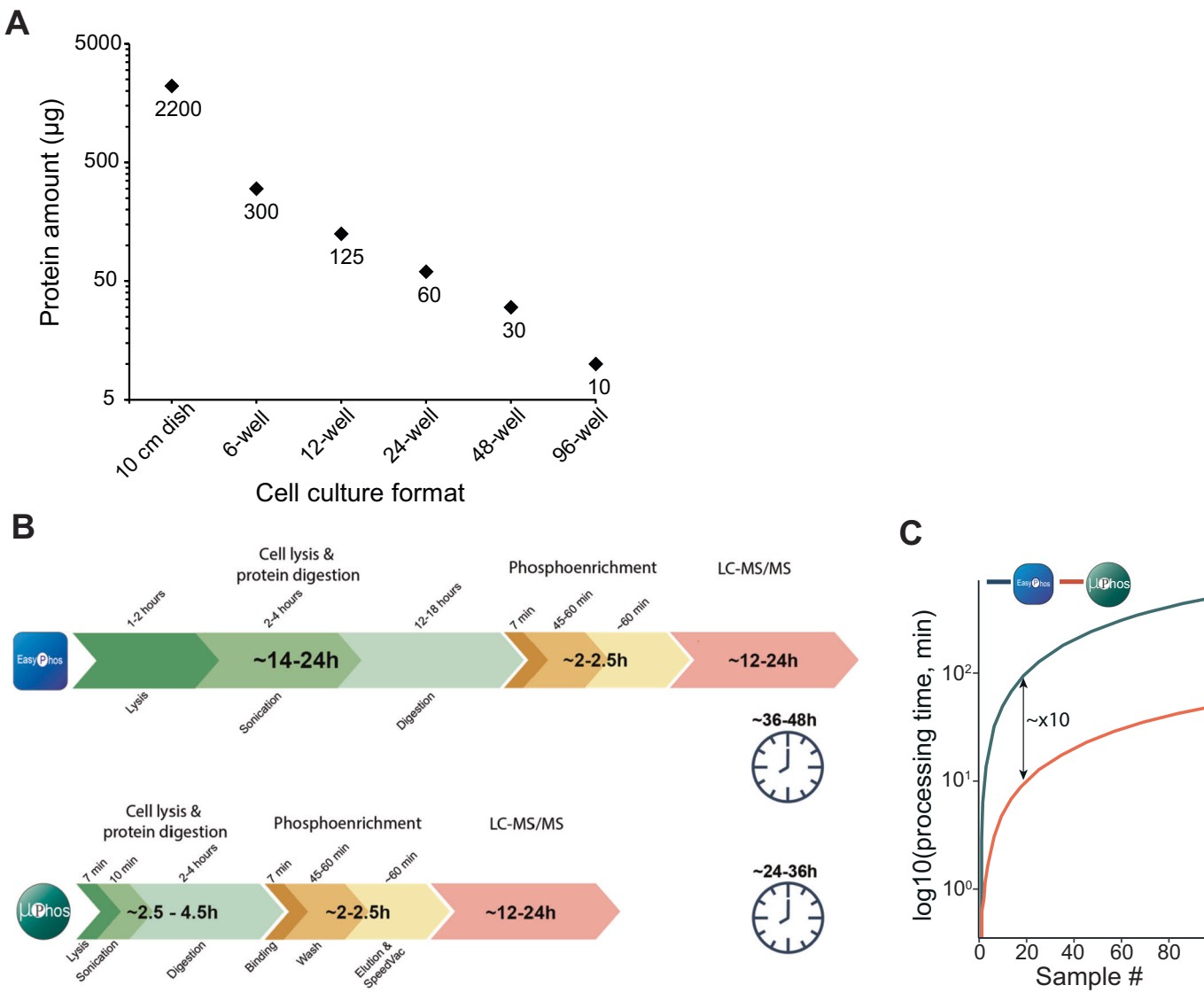

**Figure EV1.  Input amount and time considerations for large-scale phosphoproteomics.**

(A) Protein input amounts (log scale) for different cell culture formats. Estimates are based on the number of HeLa cells at confluency and assuming on average 250 pg protein per cell. Data source: https://www.thermofisher.com/de/de/home/references/gibco-cell-culture-basics/cell-culture-protocols/cell-culture-useful-numbers.html (last accessed March 13, 2024). (B) Time scale of sample handling with the µPhos protocol compared to EasyPhos. (C) Estimation of the cumulative time-per-sample for the pre-digestion processing steps calculated from the average of 96 samples.

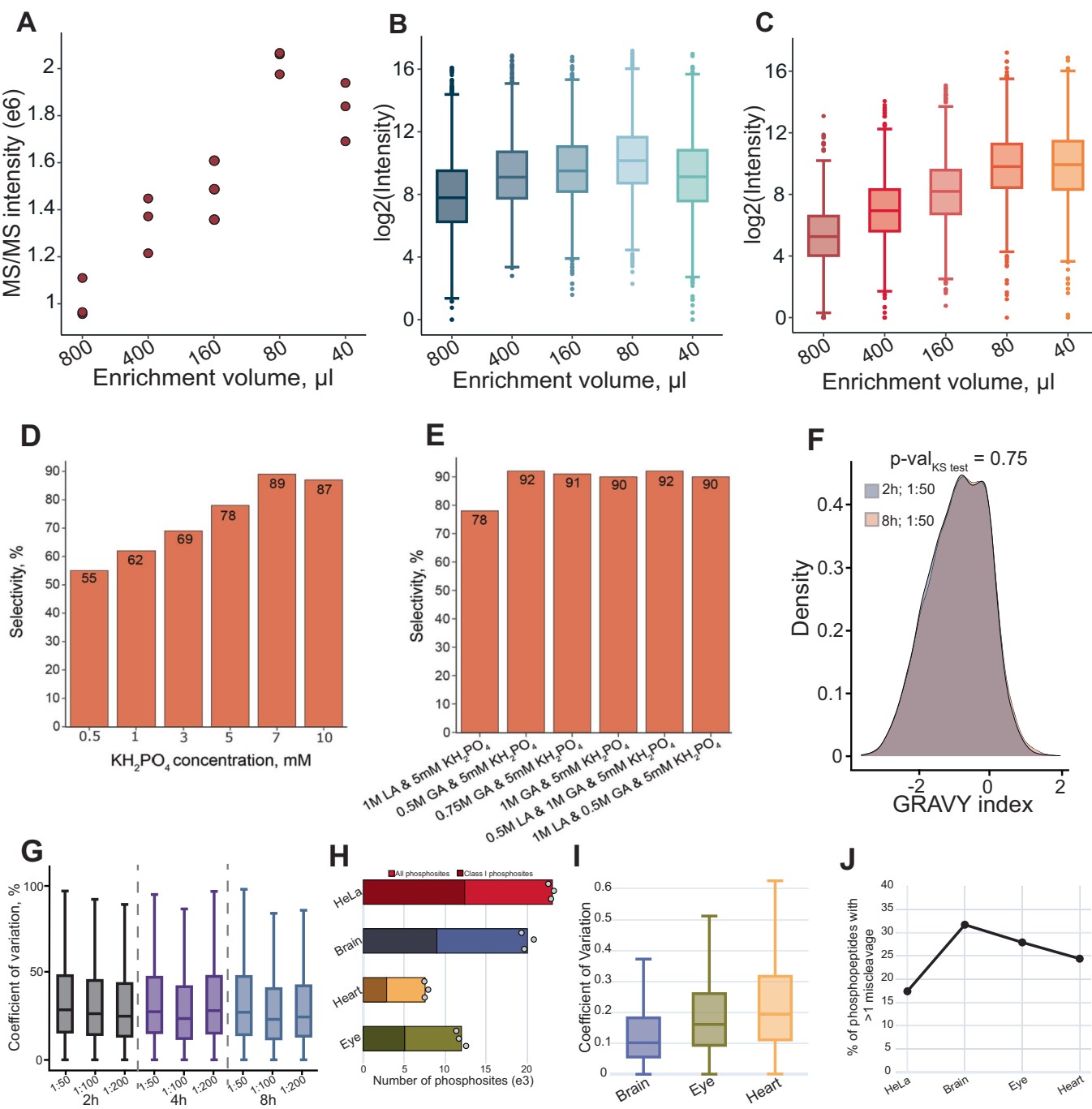

**Figure EV2. Systematic optimization of experimental parameters for µPhos.**

(**A**) Median raw fragment ion intensity in three replicate injections as a function of enrichment volume. (**B**) Logarithmized intensity of phosphopeptides for phosphopeptide enrichments with decreasing volume ($n = 3$). The box depicts the interquartile range with the central band representing the median value of the dataset. The whiskers represent the furthest datapoint within 1.5 times the interquartile range. (**C**) Same as (**B**), but for unmodified peptides ($n = 3$). The box depicts the interquartile range with the central band representing the median value of the dataset. The whiskers represent the furthest datapoint within 1.5 times the interquartile range. (**D**) Selectivity of phosphoenrichment as a function of increased concentration of monopotassium phosphate ($n = 1$). (**E**) Same as (**D**) but for combination of selectivity agents ($n = 1$). (**F**) Overlay of the GRAVY hydrophobicity index of detected phosphopeptides after either 2 h or 8 h digestion ($n = 3$). The significance of distribution overlay was determined using the Kolmogorov–Smirnov test. (**G**) Precision of label-free phosphopeptide quantification in workflow replicates ($n = 3$) for the conditions in Fig. 2D. The box depicts the interquartile range with the central band representing the median value of the dataset. The whiskers represent the furthest datapoint within 1.5 times the interquartile range. (**H**) Number of identified unique phosphosites (light colors) and Class 1 phosphosites (darker colors), enriched from 20 µg of mouse brain, heart, and eye tissue lysates and HeLa cell lysate ($n = 3$). (**I**) Precision of label-free phosphopeptide quantification in workflow replicates for the tissues in (**H**) ($n = 3$). The box depicts the interquartile range with the central band representing the median value of the dataset. The whiskers represent the furthest datapoint within 1.5 times the interquartile range. (**J**) Percent of phosphopeptides with ≥1 missed cleavage site for samples in (**H**).

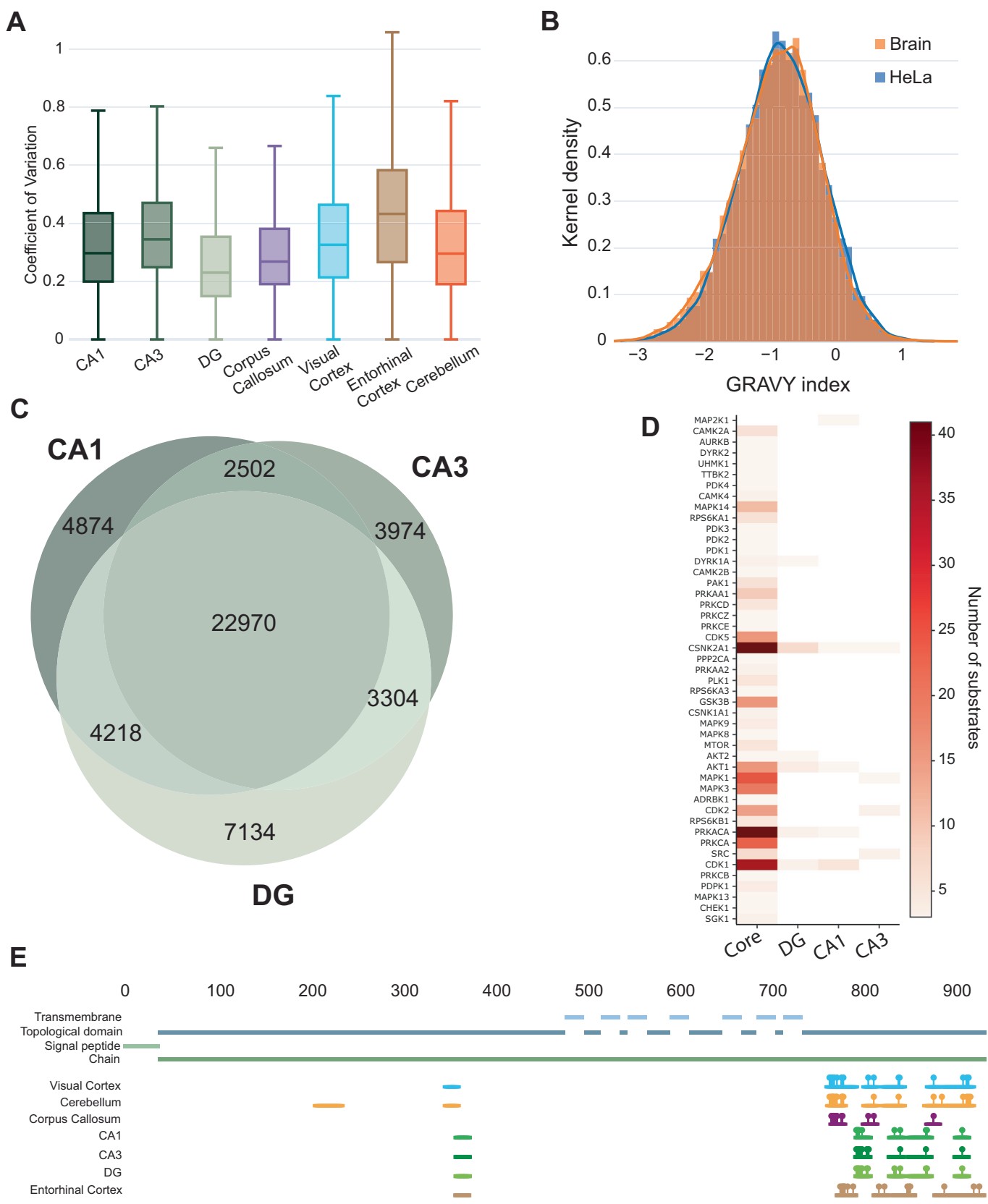

◄ **Figure EV3. Analysis of phosphoproteomes of anatomical regions in the mouse brain.**

(A) Intrareplicate quantification precision of all analyzed mouse brain regions ($n = 3$). The box depicts the interquartile range with the central band representing the median value of the dataset. The whiskers represent the furthest datapoint within 1.5 times the interquartile range. (B) Overlay of the GRAVY hydrophobicity index of phosphopeptides, detected in mouse brain and HeLa samples. (C) Overlapping phosphopeptide identifications from subregions of the murine hippocampus. (D) Number of known substrates for selected kinases in overlapping ('core') and 'subregion-specific' phosphosites from (C). (E) AlphaMap(Voytik et al, 2022) visualization of identified phosphorylation sites on the Gabbr2 receptor in different samples.

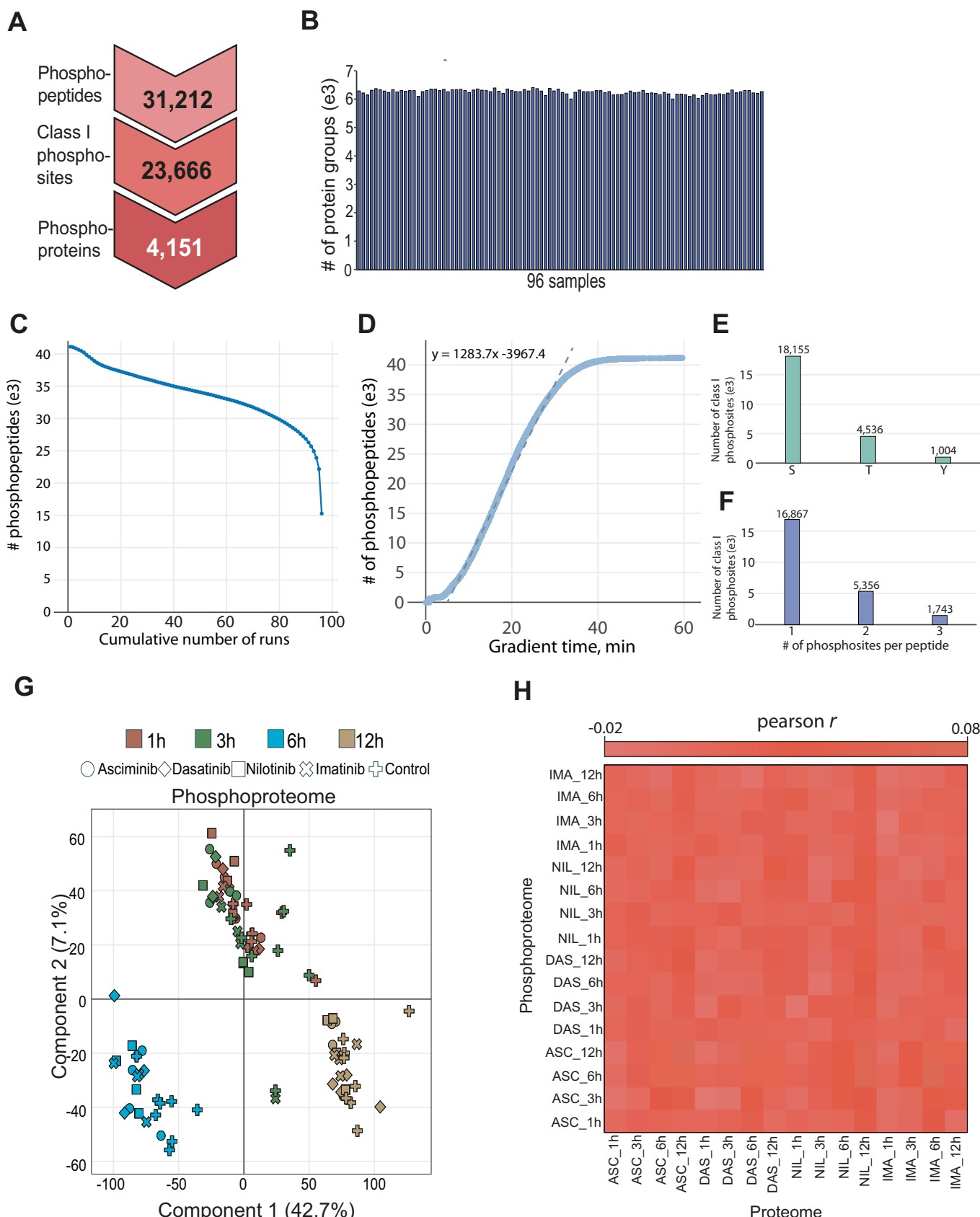

◄    **Figure EV4.   Extended analysis of drug response signatures with µPhos.**

(A) Number of identified phosphosites, class 1 phosphosites and phosphorylated protein groups. (B) Number of identified protein groups across 96 samples. (C) Cumulative number of phosphopeptides identified in *n* out of 96 experiments. (D) Number of identified phosphopeptides as a function of the chromatographic gradient. For illustration, the slope in the active part of the gradient is determined by linear regression. (E) Relative number of phosphorylated serine, threonine, and tyrosine sites in our data. (F) Relative number of singly, doubly, and triply-phosphorylated peptides in our data. (G) Principal component analysis of the phosphoproteome samples. (H) Pairwise Pearson correlation analysis of fold-changes in protein and phosphoprotein levels relative to controls.

                                    

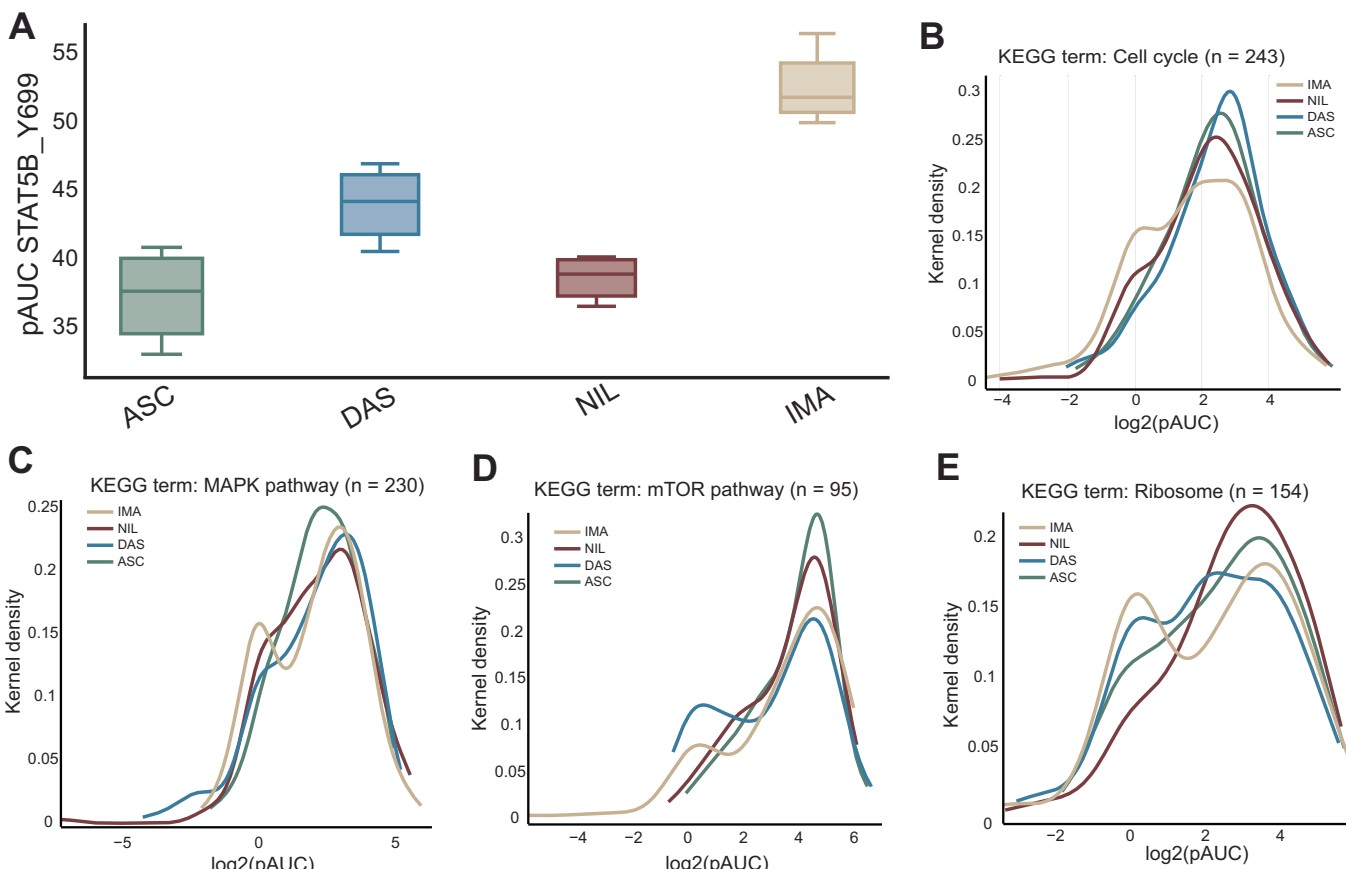

**Figure EV5. Analysis of pAUC values for selected phosphorylation sites and signaling pathways.**

(A) pAUC values of STAT5B Y699 in response to tyrosine kinase inhibition with different drugs ($n = 4$). The box depicts the interquartile range with the central band representing the median value of the dataset. The whiskers represent the furthest datapoint within 1.5 times the interquartile range. (B) Density distribution of pAUC values for phosphosites associated with the KEGG term "cell cycle". (C) Same as (B), but for the KEGG term 'MAPK pathway'. (D) Same as (B), but for the KEGG term 'mTOR pathway'. (E) Same as (B), but for the KEGG term 'Ribosome'.

