## [Peer Review File · Molecular Systems Biology]

μPhos: a scalable and sensitive platform for high-dimensional phosphoproteomics

Denys Oliinyk, Andreas Will, Felix Schneidmadel, Maximilian Böhme, Jenny Rinke, Andreas Hochhaus, Thomas Ernst, Nina Hahn, Christian Geis, Markus Lubeck, Oliver Raether, Sean Humphrey, and Florian Meier

Corresponding author(s): Florian Meier (Florian.Meier@med.uni-jena.de) , Sean Humphrey (sean.humphrey@mcri.edu.au)

Review Timeline:

Submission Date:	25th Aug 23
Editorial Decision:	16th Oct 23
Appeal Received:	3rd Nov 23
Editorial Decision:	10th Nov 23
Revision Received:	26th Mar 24
Editorial Decision:	30th Apr 24
Revision Received:	24th May 24
Editorial Decision:	4th Jun 24
Revision Received:	6th Jun 24
Accepted:	11th Jun 24

Editor: Poonam Bheda

Transaction Report:

16th Oct 2023

RE: Manuscript MSB-2023-11964, μ Phos: a scalable and sensitive platform for functional phosphoproteomics

Dear Prof Meier,

Thank you again for submitting your work to Molecular Systems Biology. We have now heard back from the three referees whom we asked to evaluate your manuscript. As you will see from the reports below, the referees raise substantial concerns on your work, which, I am afraid to say, preclude its publication in Molecular Systems Biology.

I am sorry that the review of your work did not result in a more favourable outcome on this occasion, but I hope that you will not be discouraged from sending your work to Molecular Systems Biology in the future.

Thank you for the opportunity to examine this work.

Yours sincerely,

Poonam Bheda, PhD
Scientific Editor
Molecular Systems Biology

Yours sincerely,

Poonam Bheda, PhD
Scientific Editor
Molecular Systems Biology

Reviewer #1:

This manuscript presented a workflow named μ Phos, which reduces the starting material of phosphoproteomic analysis to 20k-100k cells. Although this study shows solid technical optimization, its concept, technical advances, and scope are not favorably compared to currently proposed methods (for example, Cai et al., Communications Biology, 2023 Jan - Ref 35, in which <1k cells were analyzed together with optimization on the MS side). In our mind, the material range that the author optimized is, in fact, a relatively easy range. Going below 1k to 10k cells would be much more challenging and of higher technical significance.

In particular, the shorter sample processing time offered by μ Phos comes with a big compromise - the water-bath sonication is enough only for some (if not all) cultured cell lines, but certainly not for other samples (e.g., tissues, cell shapes, or plant cells). Similarly, the 4-hour trypsin digestion might not be enough for many other samples. In contrast, many cultured cell lines practically do not have an issue of limit amount. This consideration further reduces our enthusiasm about this work.

Also, the final peptide desalting step seems to be done still in the individual tips (Figure 1), whereas the sample collection from the human and cells as the first step in Figure IV was not drawn. The whole workflow is not a "single-pot" protocol as claimed.

The author indeed achieved higher phosphopeptide numbers than in previous workflows published. However, it is not clear whether this is due to a much better MS machine being used.

Finally, the authors only minimally analyzed the perturbation experiment of TKI inhibitors. No biological mechanisms regarding "drug-specific" phosphoproteome signatures were learned, and no verification experiments were performed. For example, it is interesting that the 6h perturbation showed a much larger effect than 12h (Figure 5d) - whether this presents a new biological insight or a proteome level variability is unclear.

Reviewer #2:

This paper describes the development of μ Phos, an easy-to-use phosphoproteomics platform that enables phosphopeptide enrichment from 96-well cell culture experiments in less than 8 hours, which is a second-generation version of EasyPhos, one of

the leading protocols for phosphoproteomics. Using highly sensitive TIMS-TOF mass spectrometry, the authors have quantified over 30,000 unique phosphopeptides in human cancer cell lines using 20 µg of starting material and demonstrated the ability to quantify approximately 6,500 phosphorylation sites from 1 µg. As an application example, time-resolved response profiling of leukemia cell lines to tyrosine kinase inhibitors was performed, demonstrating that the analytical method has sufficient depth to cover major signaling pathways.

In the field of phosphoproteomics, there are many variations in phosphopeptide enrichment methods in addition to numerous protein extraction and protein digestion methods. The comparison results also differ from one research group to another, and there is no gold standard method that is universally accepted. However, even under such circumstances, the EasyPhos method is one of the most widely recognized methods because it is well plate-based and consists of a simple workflow. The µPhos method seems to be based on the same concept, and the protocol for smaller sample volumes is very useful for researchers in this field and is worth publishing. Therefore, the method must consist of as universal a set of steps as possible and present enough information so that anyone can easily reproduce it.

Regarding the data analysis, freely available software such as DIA-NN should be used instead of commercial Spectronaut. It is known that Spectronaut provides a much higher identification number for DIA-PASEF-phosphoproteome data than DIA-NN, as a recent paper from Mann lab reported. At least the authors should use both software to show the "true" performance which can be tried by everybody in this field.

Likewise, information about the reagents used must be accurate. For instance, Empore is a product brand, not the name of the manufacturing and marketing company.

Reviewer #3:

This is well-written manuscript by Oliinyk et al., reporting a method advance of phosphoproteomics for robust handling and analysis of samples with small quantity of proteins. The overall results presented are indeed quite impressive with >30,000 phosphopeptides quantified from as low as 20 µg starting materials. The utility of the workflow was further demonstrated in an application of time-resolved experiments of cells in response to tyrosine kinase inhibitors. Nevertheless, there are several limitations of this work, which prevents me to recommend for publication in its current format.

Main:

1. Despite the impressive results being results, the current work still appears to be a rather incremental advance over the authors' own previous work on EasyPhos platform (ref. 38). Conceptually, the current workflow is an integration of one-pot processing coupled with more advanced MS instrumentation. The improvement with one-pot processing (small volume) for phosphoproteomics has already been reported (PMID: 36653408).
2. It is unclear how much the improvement observed was resulting from the improved protocol or newer instrumentation. For example, in the authors' previous work, they reported the quantification of ~10,000 phosphopeptides from ~25 µg proteins. It will be great if the authors can report the coverage using different instrumentation such as HFX vs TIMS-TOF, DIA vs. DDA. These data will be great for the broad readership. After all, the TIMS-TOF platform is still not widely available for most MS labs.
3. The data presented for the final application of time resolved phosphoproteomics were insufficient. As a reviewer, I was not able to examine the data quality from the presented information. I would suggest that the authors to make the following data available: a) the processed phosphor abundance values and associated statistics as supplementary data in spreadsheets; 2) heatmaps or volcano plots to illustrate the inhibitor responses as well as reproducibility across biological replicates.

Minor:

1. The color coding in Figure 6A is not apparent. May consider alternatives such as mini bar graphs.
2. Error bars in Figure 6B might be helpful.
3. It will be good to confirm the amounts of proteins for 50,000 cells in the final experiments.

** As a service to authors, EMBO Press offers the possibility to directly transfer declined manuscripts to another EMBO Press title or to the open access journal Life Science Alliance launched in partnership between EMBO Press, Rockefeller University Press and Cold Spring Harbor Laboratory Press. The full manuscript and if applicable, reviewers' reports, are automatically sent to the receiving journal to allow for fast handling and a prompt decision on your manuscript. For more details of this service, and to transfer your manuscript please click on Link Not Available. **

Point-by-point response**Reviewer #1:**

This manuscript presented a workflow named *μPhos*, which reduces the starting material of phosphoproteomic analysis to 20k-100k cells. Although this study shows solid technical optimization, its concept, technical advances, and scope are not favorably compared to currently proposed methods (for example, Cai et al., *Communications Biology*, 2023 Jan - Ref 35, in which <1k cells were analyzed together with optimization on the MS side). In our mind, the material range that the author optimized is, in fact, a relatively easy range. Going below 1k to 10k cells would be much more challenging and of higher technical significance.

We appreciate the reviewer's positive remark on the technical quality of our work. With all due respect, we strongly disagree with the idea that sensitivity is the only worthwhile measure of significance. On the contrary, we are convinced that, from a systems biology perspective, a protocol optimized for 20k-100k cells can have a much higher value than one for 1k-10k cells. **This is because the former enables, for the first time, systematic perturbation experiments at scale, opening up for a wide range of applications in cell signaling research.** The fact that such studies are currently performed with >200 μg of protein material (and therefore limited to just a few conditions) directly contradicts the reviewer's statement that this is "a relatively easy range" (which we find highly disrespectful considering it took a whole team about 2 years to get this protocol to its current level of performance). ***μPhos* addresses these challenges by parallel and direct processing from cell cultures to enriched phosphopeptides, representing an overall 10-fold improvement in a highly reproducible and easily accessible format.** An even higher sensitivity, while technically appealing, could have a much narrower range of applications and typically requires compromises in accessibility and scalability. For example, in our hands, we found tip-based protocols (such as Tsai et al.) prone to clogging, which is a serious limitation for up-scaling. Nevertheless, and despite the fact that the Tsai et al. protocol was specifically developed for very low cell counts, label-free *μPhos* outperforms the reported number of phosphopeptides **by a factor of two to three** in the 1-10 μg range in less analysis time.

In the revised version of our manuscript, we have rephrased the introduction and discussion to better anchor our work in this context and to highlight the conceptual novelty of our work.

In particular, the shorter sample processing time offered by *μPhos* comes with a big compromise - the water-bath sonication is enough only for some (if not all) cultured cell lines, but certainly not for other samples (e.g., tissues, cell shapes, or plant cells).

The reviewer is correct that a distinct feature of *μPhos* is the parallel lysis of cells in 96-well format. While the vast majority of cell signaling studies is performed with cell line models, we agree that other sample types might require harsher conditions for efficient lysis. A key advantage of *μPhos* is that it is compatible with a wide range of 96-well plates (Suppl. Fig. 3d-f). For this reason, water-bath sonication can be readily complemented with common 96-well plate higher-energy sonication or bead milling.

To address this point, we will test *μPhos* for different tissues types in the course of the revision.

Similarly, the 4-hour trypsin digestion might not be enough for many other samples.

In our hands, we found the 4-hour digestion sufficient for the tested range of input amounts as well as in the >100 phosphoproteomics experiments we have performed since then. In the course of the revision, we will perform additional experiments to revisit this point for different sample types.

In contrast, many cultured cell lines practically do not have an issue of limit amount. This consideration further reduces our enthusiasm about this work.

Disregarding the general advantages of less input material, this consideration is certainly justified for conventional phosphoproteomics experiments that aim to identify differentially abundant phosphosites in a low number of conditions. However, from a systems biology perspective, this approach is fundamentally limited and we and others advocate higher-dimensional experiments to deconvolute complex signaling networks (see for example Leutert *et al.*, Nat. Struc. Mol. Biol. 2023; Zecha *et al.*, Science 2023; Needham *et al.*, Sci. Signal. 2019). Ultimately, this requires moving from flask cultures to multi-well formats, *i.e.* from high to low starting amounts (Fig. R1). **The key conceptual advance of μ Phos is that it enables this novel type of phosphoproteomics experiments and makes large-scale phosphoproteomics accessible.**

Figure R1. Protein input amounts (log scale) for different cell culture formats. Estimates are based on the number of cells at confluency, assuming on average 250 pg protein per cell. State-of-the-art protocols for phosphoproteomics start from ≥ 200 μ g, while μ Phos achieves similar cover from 10x lower input material available from 96-well plates.

We hope that resolving this misunderstanding will rekindle the reviewer's enthusiasm and we will rephrase the text to convey this point more clearly in the revised version of our manuscript.

Also, the final peptide desalting step seems to be done still in the individual tips (Figure 1), whereas the sample collection from the human and cells as the first step in Figure IV was not drawn. The whole workflow is not a "single-pot" protocol as claimed.

The reviewer is correct that we desalted each sample on an individual StageTip, which can be performed for up to 384 samples in parallel with 3D-printed centrifuge adapters.

In fact, a key innovation of μ Phos is that it eliminates the need to collect and transfer cells or lysates in the first step. Instead, the cells can be processed in the very same vial and in parallel, simplifying the sample handling and minimizing losses. We will revise Figure 1 to clarify this point.

Unfortunately, Figure 4 does not show a workflow schematic and we are unable to locate the claim of a "whole-workflow single-pot protocol" the reviewer is referring to.

The author indeed achieved higher phosphopeptide numbers than in previous workflows published. However, it is not clear whether this is due to a much better MS machine being used.

We thank the reviewer for acknowledging the superior performance of our workflow. In fact, we performed a series of experiments to carefully disentangle the contributions of μ Phos from the accompanying advances in MS instrumentation and data processing. The key advances over the EasyPhos protocol are discussed and shown in Figure 2. **We think that the underlying cause of this confusion is that we labeled the reference protocol (EasyPhos) with '800 μ L' rather than 'EasyPhos'.** We will clarify the axis labeling in the revised version of our manuscript. However, comparing 800 μ L (EasyPhos) enrichment volume vs. 80 μ L (μ Phos) results in:

- **2-fold** increase in the summed MS/MS intensity (Fig. 2a)
- **4-fold** increase in phosphopeptide signal (Fig. 2d)
- **>2-fold** increase in the number of identified phosphopeptides from 20 μ g (Fig. 2b)
- **>3-fold** increase in the number of identified phosphopeptides from 20 μ g with the optimized protocol (Fig. 4a, new Supplementary Fig.)

These are substantial improvements. Importantly, all comparisons have been acquired on the very same MS platform, i.e. we expect them to be **independent from the MS instrument**. To add further evidence in the course of the revision, we will perform additional head-to-head comparisons for varying starting amounts.

Finally, the authors only minimally analyzed the perturbation experiment of TKI inhibitors. No biological mechanisms regarding "drug-specific" phosphoproteome signatures were learned, and no verification experiments were performed. For example, it is interesting that the 6h perturbation showed a much larger effect than 12h (Figure 5d) - whether this presents a new biological insight or a proteome level variability is unclear.

We agree with the reviewer that the scope and quality of the dataset invites even more in-depth analysis. At the same time, we note that the primary objective of this experiment was to demonstrate its technical feasibility for a well-described biological model with a panel of drugs that have a clearly defined therapeutic target (BCR::ABL1). The data presented in Figures 5 and 6 validate the unprecedented scalability (analyzing 96 samples in one week, requiring <10 M cells in total) as well as the accuracy (inhibition of downstream targets) of μ Phos.

We are open to discussing with the editors if and to what extent further analysis and validation experiments would add value to this *Methods* article for readers of *MSB*. Please see also our response to Reviewer #3, point 3.

Reviewer #2:

This paper describes the development of μ Phos, an easy-to-use phosphoproteomics platform that enables phosphopeptide enrichment from 96-well cell culture experiments in less than 8 hours, which is a second-generation version of EasyPhos, one of the leading protocols for phosphoproteomics. Using highly sensitive TIMS-TOF mass spectrometry, the authors have quantified over 30,000 unique phosphopeptides in human cancer cell lines using 20 μ g of starting material and demonstrated the ability to quantify approximately 6,500 phosphorylation sites from 1 μ g. As an application example, time-resolved response profiling of leukemia cell lines to tyrosine kinase inhibitors was performed, demonstrating that the analytical method has sufficient depth to cover major signaling pathways.

Thank you for the accurate summary of our work.

In the field of phosphoproteomics, there are many variations in phosphopeptide enrichment methods in addition to numerous protein extraction and protein digestion methods. The comparison results also differ from one research group to another, and there is no gold standard method that is universally accepted. However, even under such circumstances, the EasyPhos method is one of the most widely recognized methods because it is well plate-based and consists of a simple workflow. The μ Phos method seems to be based on the same concept, and the protocol for smaller sample volumes is very useful for researchers in this field and is worth publishing. Therefore, the method must consist of as universal a set of steps as possible and present enough information so that anyone can easily reproduce it.

We agree with the reviewer's perspective on the field and the key advantages of a user-friendly, well plate-based format. We highly appreciate the clear recommendation to publish our work. We believe that the Methods section of MSB is the ideal venue for this as it allows sufficient detail in a structured format that can be easily reproduced by others, as suggested by the reviewer.

Regarding the data analysis, freely available software such as DIA-NN should be used instead of commercial Spectronaut. It is known that Spectronaut provides a much higher identification number for DIA-PASEF-phosphoproteome data than DIA-NN, as a recent paper from Mann lab reported. At least the authors should use both software to show the "true" performance which can be tried by everybody in this field.

The reviewer is correct that, in addition to a variety of sample preparation protocols, data processing is subject to ongoing research in phosphoproteomics. In particular, there is no consensus on a preferred computational pipeline (quote from the cited Mann paper: "*In the context of our study, we decided to continue with the more extensive Spectronaut results, as they appeared to still correctly represent the regulation in the EGFR signaling experiment described later.*", doi: 10.1016/j.mcpro.2022.100279). For this reason, we agree with the reviewer and following the suggestion to re-analyzed key experiments with DIA-NN (Fig. 4b). This yielded very similar results for input amounts of 1 μ g, and up to ~30% difference in the number of localized phosphosites for 20 μ g input (Fig. R2). These results are in line with a more conservative localization scoring in DIA-NN. We will provide a more detailed analysis in

the course of the revision. Importantly, we already planned to make all raw data publicly available for others to reanalyze the data with their preferred software.

Figure R2. Number of identified class I phosphosites (localization score >0.75) using two alternative software packages (DIA-NN and Spectronaut (SN)) as a function of input amount of HeLa cell lysate.

Likewise, information about the reagents used must be accurate. For instance, Empore is a product brand, not the name of the manufacturing and marketing company.

Thank you for spotting this. We corrected the information in our revised manuscript and will carefully check all other items for accuracy.

Reviewer #3:

This is well-written manuscript by Oliinyk et al., reporting a method advance of phosphoproteomics for robust handling and analysis of samples with small quantity of proteins. The overall results presented are indeed quite impressive with >30,000 phosphopeptides quantified from as low as 20 μ g starting materials. The utility of the workflow was further demonstrated in an application of time-resolved experiments of cells in response to tyrosine kinase inhibitors. Nevertheless, there are several limitations of this work, which prevents me to recommend for publication in its current format.

We thank the reviewer for their encouraging summary of our work. As part of the revision, we have performed additional experiments and analysis to address the remaining concerns.

Main:

1. Despite the impressive results being results, the current work still appears to be a rather incremental advance over the authors' own previous work on EasyPhos platform (ref. 38). Conceptually, the current workflow is an integration of one-pot processing coupled with more advanced MS instrumentation. The improvement with one-pot processing (small volume) for phosphoproteomics has already been reported (PMID: 36653408).

We appreciate the reviewer's kind words about our work. To clarify the key differences to EasyPhos, we added a new Supplementary Table to our revised manuscript.

Supplementary Table. Comparison of EasyPhos and μ Phos.

Feature	EasyPhos [Humphrey et al.]	μ Phos [this study]
Phosphopeptide enrichment	TiO ₂ beads	TiO ₂ beads
Recommended input amount	200 μ g	20 μ g
Enrichment volume	800 μ L	80 μ L
Plate format	96 deep well	96-well flat-bottom, V-shape, U-shaped, deep well
Compatible lysis buffer	SDC	SDC, ACN, TFE

As discussed in our response to Reviewer #1, we think that the perceived 'incremental advance' is due to the fact that we labeled the reference protocol (EasyPhos) with '800 μ L' rather than 'EasyPhos'. Comparing 800 μ L (EasyPhos) enrichment volume vs. 80 μ L (μ Phos) **on the very same LC-MS platform** results in:

- **2-fold** increase in the summed MS/MS intensity (Fig. 2a)
- **4-fold** increase in phosphopeptide signal (Fig. 2d)
- **>2-fold** increase in the number of identified phosphopeptides from 20 μ g (Fig. 2b)
- **>3-fold** increase in the number of identified phosphopeptides from 20 μ g with the optimized protocol (Fig. 4a, new Supplementary Fig.)

Furthermore, we disagree that only the first paper on such a topic can contain any novelty, for example there is now a large number of high-profile studies in the single-cell proteomics field that all work on the common theme of one-pot/small volume sample processing. On the

contrary, we are of the opinion that the success of Tsai et al. (ref. 34, PMID: 36653408) and μ Phos warrants further studies in this direction.

2. It is unclear how much the improvement observed was resulting from the improved protocol or newer instrumentation. For example, in the authors' previous work, they reported the quantification of ~10,000 phosphopeptides from ~25 μ g proteins. It will be great if the authors can report the coverage using different instrumentation such as HFX vs TIMS-TOF, DIA vs. DDA. These data will be great for the broad readership. After all, the TIMS-TOF platform is still not widely available for most MS labs.

Please see directly above and our response to Reviewer #1. All comparisons above have been acquired on the very same MS platform, i.e. we expect them to be **independent from the MS instrument**. In the course of the revision we will add further experiments for different starting amounts. We believe that this is the most appropriate approach to benchmark a new protocol, and it would in fact not be feasible for us to acquire data on a different instrument platform.

3. The data presented for the final application of time resolved phosphoproteomics were insufficient. As a reviewer, I was not able to examine the data quality from the presented information. I would suggest that the authors to make the following data available: a) the processed phosphor abundance values and associated statistics as supplementary data in spreadsheets; 2) heatmaps or volcano plots to illustrate the inhibitor responses as well as reproducibility across biological replicates.

Thank you. Following the reviewer's suggestion, we will add new Supplementary Tables to the revised manuscript. In addition, we will present a more in-depth analysis of the dataset, which we have already started (see Figures R3 and R4 on the next page).

We are open to discussing with the editors to what extent further analysis would add value to this *Methods* article for readers of *MSB*.

Figure R3. Pairwise Volcano-plot analysis of our high-dimensional kinase inhibitor analysis of a murine chronic myeloid leukemia model. Left to right: time-course; Top to bottom: kinase inhibitors.

Figure R4. Heatmap visualization of inferred kinase activity in response to four tyrosine kinase inhibitors.

Minor:

1. The color coding in Figure 6A is not apparent. May consider alternatives such as mini bar graphs.

We agree that mini bar graphs can improve the readability of the Figure.

2. Error bars in Figure 6B might be helpful.

Thank you, we added error bars to Fig. 6B of the revised manuscript.

3. It will be good to confirm the amounts of proteins for 50,000 cells in the final experiments.

Thank you, we added this information to the revised manuscript.

10th Nov 2023

RE: Manuscript MSB-2023-11964R-Q, "μPhos: a scalable and sensitive platform for functional phosphoproteomics"

Dear Dr. Meier,

Thank you for your correspondence regarding our editorial decision on your manuscript MSB-2023-11964 "μPhos: a scalable and sensitive platform for functional phosphoproteomics". I have now had the chance to read once again the manuscript and the reviewers' comments. I have also read the points raised in your appeal letter and your preliminary point-by-point response and discussed them with the team. As I will explain below we would not be opposed to considering a revised and extended manuscript addressing the issues raised by the reviewers.

During the review of the study, all three reviewers mentioned that the methodological novelty seems somewhat limited. Given that there are other optimized protocols for low sample size for phosphoproteomics, the reviewers mentioned that additional comparisons for sample processing, MS instrumentation, and software for analysis would be required. During our cross-consultation session between reviewers and editors, all three reviewers agreed that even though there is value in the technical improvement, the advance itself seems rather incremental. In addition, both Reviewers 1 and 3 pointed out the limited application of uPhos for deriving new biological insights. The limited level of biological insight provided by the study represents an important limitation, given that the novelty from a methodological point of view seems somewhat limited. On balance, the reviewers' comments indicated that the initial version of the study did not provide the kind of decisive advance with a demonstrated potential to derive new biology.

We think that your responses to the reviewer critiques especially in terms of MS instrumentation agnostic improvements in the protocol seem reasonable and we understand that several of the issues raised may have been due to misunderstandings and could be clarified. We also think that the planned experiments and analyses to demonstrate further the utility of the method under various conditions and with readily available MS instruments and software seem potentially promising for addressing the related concerns of the reviewers. As indicated above, in light of the reviewers' concerns on the limited methodological novelty, an important point that would need to be addressed would be increasing the level of biological insight provided by the study. As such, it would be important to extend the biological insight from the TKI perturbation experiments for further consideration at MSB. We appreciate that as you indicate in your letter and response, you would be willing to extend the study along those lines. Overall, we think that the proposed clarifications and additional analyses seem potentially promising for addressing the reviewers' concerns. As such, and given that the reviewers did have positive words about the potential relevance of the study for the proteomics field, we would invite you to submit an extended and revised manuscript, which will be sent back to the reviewers so that they can assess if their concerns have been satisfactorily addressed.

As you probably understand, and given that addressing some of the more substantial issues raised involves analyses with unclear outcome, we can give no guarantee about the eventual acceptability of the study. If you decide to follow this course, we would ask you to please submit as a new submission but nevertheless enclose a point-by-point response to the comments raised by the reviewers.

I hope that the comments above will be helpful in deciding how to further proceed with the study.

With kind regards,
Poonam

Poonam Bheda, PhD
Scientific Editor
Molecular Systems Biology

** As a service to authors, EMBO Press offers the possibility to directly transfer declined manuscripts to another EMBO Press title or to the open access journal Life Science Alliance launched in partnership between EMBO Press, Rockefeller University Press and Cold Spring Harbor Laboratory Press. The full manuscript and if applicable, reviewers' reports, are automatically sent to the receiving journal to allow for fast handling and a prompt decision on your manuscript. For more details of this service, and to transfer your manuscript please click on Link Not Available. **

Manuscript MSB-2023-11964, *μPhos*: a scalable and sensitive platform for high-dimensional phosphoproteomics

Point-by-point response

We thank the reviewers and editors for their thorough assessment of our work. Their detailed and constructive comments have helped us to clarify important aspects of our work and to extend the scope of our manuscript significantly.

The reviewers were impressed by our results and remarked positively on the quality of our study, highlighting the potential impact of robust, low-input phosphoproteomics. A common concern was that aspects of our *μPhos* workflow were not entirely new, referring to previous work by one of us (Humphrey *et al.*, *Nat. Protocols* 2018) and a more recent study by Tsai *et al.* (*Comms. Biol.* 2023). We have carefully revised the text to better contextualize our manuscript with respect to previous work in the field, and to ensure that key contributions are acknowledged. As noted by Reviewers #1 and #3, *μPhos* outperforms existing protocols in terms of phosphoproteome coverage, and we have now clarified that the four-fold increase in signal over its predecessor is independent of MS instrumentation, but a result of minimizing volumes and transfer steps. That said, we believe that the conceptual advance of our study goes beyond a mere technical optimization for low input amounts. This is because *μPhos* achieves excellent sensitivity without compromising robustness and accessibility, thereby enabling a shift in cell signaling research from classical experimental designs with few conditions to high-dimensional, systematic perturbation experiments at scale.

The reviewers and editors also suggested that the manuscript could be strengthened by providing additional evidence of the potential to derive new biological insights using *μPhos*. We therefore performed additional experiments to validate results from the drug response experiment and devised a new strategy to analyze multi-inhibitor time-course data. This highlighted phosphosites that were inhibited with low inter-drug variability in the BCR::ABL1 signaling pathway, as well as nuanced distinctions among the four small molecule inhibitors.

Finally, the reviewers questioned the utility of *μPhos* beyond cell culture models. To explore this, we evaluated it with complex tissue proteomes and also employed it to map, for the first time, spatially-resolved phosphoproteomes of small anatomical regions in the mouse brain. This revealed a remarkable heterogeneity in the phosphoproteome of subregions of the hippocampal formation and quantified distinct phosphorylation patterns on individual G-protein coupled receptors, which we believe will be of great interest to the wider GPCR research community and the field of neurobiology as a whole.

To maximize transparency and facilitate data access, we added new Supplementary Tables with phosphosite-level data and deposited all our raw data in the PRIDE/ProteomeXchange repository.

Below we have addressed each reviewers' comments in detail in a point-by-point response. To facilitate review, we provide a version of the manuscript with all major changes highlighted. We hope that the reviewers and editors agree that these changes have strengthened our manuscript and find the revised version suitable for publication as a *Methods* article in *Molecular Systems Biology*.

Reviewer #1:

This manuscript presented a workflow named *μPhos*, which reduces the starting material of phosphoproteomic analysis to 20k-100k cells. Although this study shows solid technical optimization, its concept, technical advances, and scope are not favorably compared to currently proposed methods (for example, Cai et al., *Communications Biology*, 2023 Jan - Ref 35, in which <1k cells were analyzed together with optimization on the MS side). In our mind, the material range that the author optimized is, in fact, a relatively easy range. Going below 1k to 10k cells would be much more challenging and of higher technical significance.

We appreciate the reviewer's positive remark on the technical quality of our work and regret that some of the advances in our original manuscript may not have been sufficiently clear. The comments helped us to highlight key conceptual and technical advances, and to extend the scope of our work by additional experiments and analyses.

We respectfully disagree with the notion that sensitivity is the only worthwhile measure of significance. In fact, we are convinced that, from a systems biology perspective, a robust and accessible phosphoproteomics workflow for 20k-100k cells can be even more valuable than one for <1k cells. This is because the former enables systematic perturbation experiments at scale, opening up a wide range of applications in cell signaling research.

Pioneering studies in this field are currently performed with >200 μg of protein material (PMID: 36926954, 37845410, 34857927). This entails high cost, complexity, and significant hands-on processing time. This contradicts the reviewer's notion that the range we optimized for is "a relatively easy range" and suggests that miniaturization and up-scaling is in fact quite challenging, involving more than simply increasing sensitivity. *μPhos* addresses these challenges by parallel and direct processing from cells to enriched phosphopeptides, representing an overall 10-fold improvement (from 200 to 20 μg input material) in a highly reproducible and accessible plate format.

In the revised manuscript, we reworded the introduction and added **Fig. EV1A** to emphasize these conceptual advances. We estimate that 96-well cell culture experiments (a reasonable target for systems biology) yields ~10 μg protein per sample. This in turn suggests diminishing returns from even higher sensitivity, especially considering that proposed alternatives typically require compromises in reproducibility, accessibility and scalability. For example, the method proposed by Tsai *et al.* involves the manufacturing of custom pipette tips with Ni-NTA silica beads and requires C₁₈ solid phase extraction before and after the phosphopeptide enrichment, while the tips are manually inserted into each other at each step (PMID: 36653408). In contrast, *μPhos* minimizes potential material losses before the phosphopeptide enrichment by performing cell lysis, digestion, and enrichment in the same reaction vessel (see revised **Fig. 1**). In addition, *μPhos* inherits the robustness and accessibility of well plate-based protocols such as *EasyPhos* that have contributed to their widespread adoption in the field, as noted by Reviewer #2. Nevertheless, and despite the fact that the Tsai *et al.* protocol was specifically developed for very low cell counts, our workflow outperforms the reported number of phosphopeptides in label-free experiments by a factor of 2-3X in the 1-10 μg input range, in less analysis time. We rephrased the discussion in our revised manuscript to better anchor our work in this context.

In particular, the shorter sample processing time offered by *μPhos* comes with a big compromise - the water-bath sonication is enough only for some (if not all) cultured cell lines, but certainly not for other samples (e.g., tissues, cell shapes, or plant cells).

The reviewer is correct that a distinct feature of *μPhos* is the parallel lysis of cells in 96-well format and we found water-bath sonication suitable for this task as initially described for cell culture samples (**Fig. 2, Figs. EV2-4**). In the course of the revision we therefore performed additional experiments to demonstrate the applicability of *μPhos* beyond cell line models. Specifically, we set out to map the phosphoproteome of small anatomical mouse brain regions (new section '**subregion-resolved phosphoproteomics of the mouse brain**', **Fig. 4, Fig. EV3**). We found *μPhos* well applicable to tissue samples, as evident from comprehensive phosphoproteome coverage and excellent reproducibility. This result is further corroborated by similar physicochemical properties of phosphopeptides detected in cell line and tissue samples (**Fig. EV3b**) and exemplified by the excellent coverage of phosphorylation sites on an integral membrane receptor protein (**Fig. 4f**), representing a class of notoriously difficult proteins.

We agree with the reviewer that higher sample amounts or other sample types might still benefit from harsher conditions for cell lysis. However, please note that a key advantage of *μPhos* is its wide compatibility with different 96-well plate formats (**Fig. Appendix S1**). Thus, while water-bath sonication is the most accessible option for many laboratories, it should be straightforward to include 96-well plate higher-energy sonication or bead milling as needed. We added this point to the discussion of our revised manuscript.

Similarly, the 4-hour trypsin digestion might not be enough for many other samples.

Thank you for raising this point. While we found the 4-hour digestion suitable for the optimized range of input amounts (**Fig. 2d**) as well as in the >100 phosphoproteomics experiments we have performed since, this comment motivated us to revisit the digestion efficiency for other sample types. In addition to the tissue samples described in response to the point just above, we analyzed diverse murine tissue lysates (brain, heart, eye). For 20 μg protein input amounts, the 4-hour trypsin/LysC digestion resulted in miscleavage rates <30% and high enrichment selectivity (>70% of identified peptides were phosphorylated). In line with our cell line results, we found our workflow highly reproducible with median coefficients of variation <20%. From these results, we concluded that the 4-hour digestion indeed provides adequate proteolysis efficiency under our conditions, even for highly complex proteomes. Interestingly, Leutert *et al.* (*Mol Sys Biol* 2019) also suggested a 3.5 h digestion time for phosphoproteome analysis of yeast samples. In either case, this parameter can be easily adjusted as needed and we point this out in the discussion of our revised manuscript.

In contrast, many cultured cell lines practically do not have an issue of limit amount. This consideration further reduces our enthusiasm about this work.

Thank you for this comment. Disregarding the general advantages of less input material such as reduced cost, this consideration is certainly justified for conventional phosphoproteomics experiments that aim to identify differentially abundant phosphosites in a small number of conditions. However, from a systems biology perspective, this approach is fundamentally

Figure R1. Protein input amounts (log scale) for different cell culture formats. Estimates are based on the number of HeLa cells at confluency and assuming on average 250 pg protein per cell.

Data source:

<https://www.thermofisher.com/de/de/home/references/gibco-cell-culture-basics/cell-culture-protocols/cell-culture-useful-numbers.html> (last accessed Mar 13 2024).

limited and we and others advocate higher-dimensional experiments to deconvolute complex signaling networks (see for example Leutert *et al.*, Nat. Struc. Mol. Biol. 2023; Zecha *et al.*, Science 2023; Needham *et al.*, Sci. Signal. 2019). Ultimately, this requires moving from flask or dish cultures to multi-well formats, *i.e.* from high to low starting amounts (**Fig. R1**, new **Fig. EV1A**). A key conceptual advance of μ Phos is that it enables high-dimensional phosphoproteomics experiments in 96-well format and makes them accessible for a large number of laboratories. To achieve this, we designed μ Phos to (i) reduce all operation volumes to ~ 100 μ L, (ii) allow plate-based processing of 96 samples in parallel with minimal transfer steps and hands-on time, and (iii) avoid reliance on specialized equipment or reagents. Note that these advances should also benefit more conventional experimental designs, for example by allowing additional replicates or controls to be included due to reduced cost and simplified handling.

In the revised version of our manuscript, we changed the title, rephrased the text and added **Fig. EV1A** to convey these points more clearly and resolve the apparent misunderstanding.

Also, the final peptide desalting step seems to be done still in the individual tips (Figure 1), whereas the sample collection from the human and cells as the first step in Figure IV was not drawn. The whole workflow is not a "single-pot" protocol as claimed.

The reviewer is correct that we desalted each sample on an individual StageTip, which can be performed for up to 384 (4x96) samples in parallel with 3D-printed centrifuge adapters or (semi-)automated as shown in numerous studies.

We do not see how our original manuscript could have been interpreted to be a "whole-workflow single-pot protocol", but we have carefully edited the text to avoid any potential misunderstanding. In fact, a key innovation of μ Phos is that it eliminates the need to collect and transfer cells or lysates in the first step. Instead, cells can be processed in the very same vessel and in parallel, simplifying sample handling and minimizing losses.

We revised **Figure 1** to clarify these points and also refer readers to the structured protocol format, which is an essential feature of *Methods* articles in *Molecular Systems Biology*.

Figure 1. Design of the *μPhos* platform.

Schematic overview of the *μPhos* workflow for high-dimensional phosphoproteomics. The protocol streamlines cell lysis, proteolytic digestion and phosphopeptide enrichment in low volumes, allowing it to be performed in a single reaction vessel in plate format with high sensitivity. Enriched phosphopeptides are eluted from TiO₂ beads and desalted with StageTips, and subsequently analyzed by LC-MS/MS.

The author indeed achieved higher phosphopeptide numbers than in previous workflows published. However, it is not clear whether this is due to a much better MS machine being used.

We thank the reviewer for the positive remark on the performance of our workflow. During the development of the protocol, we performed a series of experiments to carefully disentangle the contributions of *μPhos* from the accompanying advances in MS instrumentation and data processing. In the revised version of our manuscript, we clarified our choice of *EasyPhos* (one of the leading protocols as noted by Reviewer #2) as the starting and reference point for our development.

The key advances over *EasyPhos* are discussed in the section “**Development of a reproducible one-day protocol in small volumes**” and shown in **Figure 2** as well as Figs. **EV1** and **EV2**. We think that the underlying cause of the confusion was that we did not clearly indicate or discuss the reference point in all cases. We clarified this in the revised version of our manuscript, for example:

“Investigating the raw intensities in more detail showed that the median phosphopeptide intensity increased 4-fold by decreasing the enrichment volume from 800 μL (i.e. EasyPhos) to 40 μL (Fig. EV2B).”

In summary, comparing 800 μL (*EasyPhos*) enrichment volume vs. 80 μL (*μPhos*) resulted in:

- 2-fold increase in the summed MS/MS intensity (**Fig. EV2a**)
- 4-fold increase in phosphopeptide signal (**Fig. EV2b**)
- 2-fold increase in the number of identified phosphopeptides from 20 μg HeLa lysate (**Fig. 2a**)
- 2-3-fold increase in the number of identified phosphosites from 20 μg HeLa lysate (**Fig. EV2h**)

These are substantial improvements over the state of the art. Importantly, all comparisons were acquired on the very same MS platform, i.e. we expect them to be independent from the MS instrument. We added this important point to the discussion of our revised manuscript.

Finally, the authors only minimally analyzed the perturbation experiment of TKI inhibitors. No

biological mechanisms regarding "drug-specific" phosphoproteome signatures were learned, and no verification experiments were performed. For example, it is interesting that the 6h perturbation showed a much larger effect than 12h (Figure 5d) - whether this presents a new biological insight or a proteome level variability is unclear.

We agree with the reviewer that our original manuscript focused on demonstrating the technical feasibility of complex experimental designs, and thereby lost sight of the biological findings to some extent. In the revised manuscript, we combined the original Figures 5 and 6 into the new **Figure 5** and supplementary figures to condense these technical aspects, leaving more space for interpretation of the resulting data. These data validate the scalability (analyzing 96 samples in one week, requiring <10 M cells in total) and accuracy (inhibition of downstream targets) of μ Phos, using a well-described biological model with a panel of drugs that have a defined therapeutic target (BCR::ABL1). In the course of the revision, we also performed additional immunoblotting experiments to verify the accuracy of our MS-based approach at this scale (**Fig. 5e**).

Moreover, we agree that the scope and quality of this dataset invites a more in-depth analysis. Specifically, our dataset contains virtually complete longitudinal profiles for 23,666 phosphosites across different drugs, providing additional information to conventional measurements of log-fold changes at a single time-point and for a single drug. To leverage the high dimensionality of this data, we devised an analysis strategy based on the time-weighted area under the curve (new section "**Time-course analysis highlights drug-specific response signatures**", new **Fig. 6**). This highlighted subtle variations (and commonalities) in the drug-specific phosphoproteome signatures, which might have remained elusive in classic experimental designs.

"Time-course analysis highlights drug-specific response signatures

*Given that BCR::ABL1 is the common therapeutic target of all four drugs in our model, we asked whether μ Phos could also dissect more subtle variations in drug response, such as those arising from potential off-target inhibition or signaling crosstalk. Our dataset contains longitudinal profiles for 23,666 phosphosites, providing additional information to conventional measurements of log-fold changes at a single time point and for a single drug. To prioritize phosphosites by the speed and duration of their response, we calculated the time-weighted area under the curve for each phosphosite ('pAUC') (**Fig. 6a**). This resulted in pAUC values ranging between 0 and 91 a.u. and, confirming the suitability of this score, we found Y699 on STAT5B consistently among the 2% most inhibited sites (pAUC >35) for all TKIs (**Fig. EV5A**). As an estimate of the overall drug efficacy, we plotted the cumulative pAUC for all ANOVA-*

Figure 6. Time-response signatures of tyrosine kinase inhibitors.

a Schematic illustration of the time-weighted area under the curve (pAUC). For each phosphosite, we determined the area under the curve and multiplied it by T_{\max}/T to weight the speed of the drug response.

b Cumulative sum of pAUC values for ANOVA-significant phosphosites as a function of their rank.

c Inter-drug variability of phosphosite pAUC values.

d Number of known PhosphoSitePlus substrates for selected kinases in core (CV ≤ 0.2) and variable (CV ≥ 0.75) phosphosites.

e Enrichment of GO terms among core and variable phosphosites. (Benjamini-Hochberg FDR < 0.02)

f Density distribution of pAUC values for phosphosites associated with the GO term "histone deacetylation".

g Phosphorylation levels of HDAC sites in response to TKI treatment.

significant phosphosites as a function of their rank order (Fig. 6b). Interestingly, out of the ~17,000 sites in our dataset, only ~3,000 accounted for 50% of the summed pAUC for each inhibitor.

To explore drug-specific phosphoproteome variations, we next defined ‘core’ phosphosites as those with consistent response patterns between drugs (intra-drug CV ≤ 0.2) in contrast to ‘variable’ phosphosites (intra-drug CV ≥ 0.75) (Fig. 6c). The core phosphosites included substrates of ABL1 (Stat5b Y699, Dok1 Y369, Cbl Y698, Crkl Y207) as well as sites phosphorylated by Cdk1, Cdk2, Gsk3b and Akt1 (Fig. 6d). Conversely, we observed several substrates of Mapk3, Prkc and Src among the variable phosphosites. Gene Ontology (GO) analysis linked the core phosphoproteome to RNA splicing, apoptosis, protein maintenance, cell growth and differentiation and enzyme activity, suggesting that these biological processes are commonly affected by BCR::ABL1 inhibitors (Fig. 6e). Interestingly, ‘chromatin organisation’, ‘DNA repair’ and related terms were enriched in the variable phosphoproteome, including regulatory phosphosites on proteins of the histone deacetylation machinery. To investigate this further, we filtered for phosphosites on proteins associated with the GOBP term ‘histone deacetylation’ and calculated the density distribution of their corresponding pAUCs (Fig. 6f). Setting it apart from all other drugs, we observed a shift towards lower effects size (i.e. lower pAUC) for imatinib. To validate this result, we repeated our analysis for GO terms related to invariable phosphosites (‘MAPK signaling pathway’, ‘mTOR signaling pathway’, ‘Cell cycle’ and ‘Ribosome’), which resulted in indistinct density distributions for all drugs (Fig. EV5B-E). At the site level, we confirmed the dephosphorylation of regulatory sites on histone deacetylase 1 (HDAC1) and other HDACs in response to asciminib, nilotinib and dasatinib, which was attenuated for imatinib in our experiment (Fig. 6g). This is an intriguing finding and, given the prospects for combination therapies with HDAC and tyrosine kinase inhibitors^{61–63} demonstrates the potential to gain relevant biological insights by combining high-dimensional perturbation experiments with μ Phos.”

To further demonstrate the potential of μ Phos in an additional application area, as mentioned above, we analyzed the phosphoproteome of small anatomical regions in the mouse brain (new section ‘**subregion-resolved phosphoproteomics of the mouse brain**’, new Fig. 4).

“We reasoned that the scalability and sensitivity of μ Phos should be advantageous beyond cell line proteomes. To illustrate this, we set out to map the phosphoproteome of small mouse brain regions. The fine-grained cellular and spatial organization of the brain underlies its diverse physiological functions, and is reflected at different molecular levels⁴⁶. However, limited sample amounts have so far constrained large-scale spatial phosphoproteomics studies to major anatomical regions, leaving finer details elusive⁴⁷. Here, we dissected seven anatomically distinct areas from six adult mice, including visual cortex (VC), entorhinal cortex (EC), corpus callosum (CC) and cerebellum (CB), as well as three subregions of the hippocampal formation (dentate gyrus (DG), Cornu ammonis 1 (CA1) and Cornu ammonis 3 (CA3)) (Fig. 4a). Tissue volumes of the latter ranged between 0.2-0.6 mm³ and by sub-sampling these regions we were able to extract only 10 to 40 μ g of protein material. Despite this low input, we identified ~70,000 unique phosphosites in total and more than 30,000 phosphosites in the hippocampus alone, mapping to 6,043 and 3,891 phosphoproteins, respectively (Fig. 4b). We found the one-day μ Phos protocol readily applicable to small tissue amounts, resulting in an excellent reproducibility across animals with median CVs of ~30% (Fig. EV3A, B). To obtain an overview of the brain’s phosphoproteome architecture, we performed a principal component analysis (Fig. 4c), which recapitulated the previously reported separation of the cerebrum regions (cortex, hippocampus) from the cerebellum⁴⁷,

while the corpus callosum formed a distinct cluster. Samples from the entorhinal cortex, the interface of the trisynaptic circuit, clustered closer to the hippocampal formation than the visual cortex and, interestingly, the CA1 and CA3 subregions of the hippocampal formation were separated from the dentate gyrus.

Figure 4. Phosphoproteome analysis of murine brain subregions.

a Schematic drawing of the analyzed anatomical regions. The inset zooms into the hippocampal formation.

b Number of identified phosphosites and Class I phosphosites (localization score > 0.75 , darker color).

c Principal component analysis, showing the first and second component.

d Pairwise Pearson correlation analysis of all samples. The inset shows a scatter plot for entorhinal cortex vs. CA3.

e Volcano plot of inferred kinase activities based on sequence recognition motifs⁴⁸. Colored points indicate kinases with FDR-corrected q -values < 0.05 and a minimum absolute \log_2 fold-change of 0.5.

f Heatmap of phosphosite abundance on *Gabbr2* across different brain regions.

The latter likely reflects differences in their cellular composition as both CA subregions are rich in glutamergic pyramidal neurons, while the DG is densely populated with granule cells. These findings were further corroborated by a pairwise Pearson correlation analysis, which revealed a decreasing correlation for developmentally distant brain regions (Fig. 4d). To investigate phosphoproteome heterogeneity in the hippocampal formation in more detail, we analyzed the DG, CA1 and CA3 samples separately (Fig. EV3C, D). Integrating our data with experimentally derived substrate sequence specificities of serine/threonine kinases⁴⁸ indicated similar basal kinase activities in CA1/CA3, and a distinct kinase activity profile for the DG (Fig. 4e). Kinases with inferred high activity (\log_2 fold-change: 0.8) included two subunits of the calcium/calmodulin-dependent protein kinase type II (*Camk2*), which is strongly associated with neurogenesis and synaptic plasticity⁴⁹ and thereby in line with the physiological function of the DG⁵⁰. The DG phosphoproteome was further enriched for

*recognition motifs of G protein-coupled receptor kinases (GRKs) – primary regulators of G protein-coupled receptor (GPCR) sensitivity. The mismatch between >800 GPCRs and only seven known GRKs led to the hypothesis of phosphorylation ‘barcodes’ on the cytosolic GPCR domain as a mechanism of modulating downstream signaling^{51–53}. This motivated us to examine whether *μPhos* is capable of quantifying phosphorylation events on GPCRs. As an example, we inspected the Gamma-aminobutyric acid type B receptor subunit 2 (*Gabbr2*), which is broadly expressed in astrocytes⁵⁴. Remarkably, we detected phosphorylation on 20 out of 34 Ser/Thr sites in the protein C-terminus, which essentially covers the entire intracellular domain of the receptor (**Fig. EV3E**) and revealed a distinct phosphorylation pattern for each brain region (**Fig. 4f**). Collectively, these results pave the way for studies of receptor phosphorylation and downstream signaling events in vivo with unprecedented detail.”*

We think that both experiments showcase the feasibility to explore uncharted terrain and generate novel biological discoveries with *μPhos*. However, with reference to the editorial guidelines, we consider the further biological validation of potentially novel regulatory mechanisms beyond the scope of a *Methods* article in *Molecular Systems Biology*.

Please see also our response to Reviewer #3, point 3.

Reviewer #2:

This paper describes the development of *μPhos*, an easy-to-use phosphoproteomics platform that enables phosphopeptide enrichment from 96-well cell culture experiments in less than 8 hours, which is a second-generation version of *EasyPhos*, one of the leading protocols for phosphoproteomics. Using highly sensitive TIMS-TOF mass spectrometry, the authors have quantified over 30,000 unique phosphopeptides in human cancer cell lines using 20 μg of starting material and demonstrated the ability to quantify approximately 6,500 phosphorylation sites from 1 μg. As an application example, time-resolved response profiling of leukemia cell lines to tyrosine kinase inhibitors was performed, demonstrating that the analytical method has sufficient depth to cover major signaling pathways.

We thank the reviewer for their accurate summary of our manuscript.

In the field of phosphoproteomics, there are many variations in phosphopeptide enrichment methods in addition to numerous protein extraction and protein digestion methods. The comparison results also differ from one research group to another, and there is no gold standard method that is universally accepted. However, even under such circumstances, the *EasyPhos* method is one of the most widely recognized methods because it is well plate-based and consists of a simple workflow. The *μPhos* method seems to be based on the same concept, and the protocol for smaller sample volumes is very useful for researchers in this field and is worth publishing. Therefore, the method must consist of as universal a set of steps as possible and present enough information so that anyone can easily reproduce it.

We agree with the reviewer's perspective on the field and the key advantages of a user-friendly, well plate-based format. We appreciate the recommendation to publish our manuscript. We believe that the *Methods* section of *Molecular Systems Biology* is the ideal venue for this as it allows sufficient detail in a structured format that can be easily reproduced by others, as requested by the reviewer.

Regarding the data analysis, freely available software such as DIA-NN should be used instead of commercial Spectronaut. It is known that Spectronaut provides a much higher identification number for DIA-PASEF-phosphoproteome data than DIA-NN, as a recent paper from Mann lab reported. At least the authors should use both software to show the "true" performance which can be tried by everybody in this field.

Thank you for raising this point. The reviewer is correct that, in addition to a variety of sample preparation protocols, data processing is subject to ongoing research in phosphoproteomics (PMID: 36609502). In particular, there is no consensus on a preferred computational pipeline for data-independent acquisition experiments (quote from the referenced paper from Mann and colleagues: "*In the context of our study, we decided to continue with the more extensive Spectronaut results, as they appeared to still correctly represent the regulation in the EGFR signaling experiment described later.*", doi: 10.1016/j.mcpro.2022.100279). In part, this is due to the challenges associated with localizing phosphorylation on specific amino acids in the sequenced peptides. Here, we filter for unique sites as described in the Methods section to avoid inflated phosphopeptide numbers due to ambiguous site localizations.

Following the reviewer's suggestion, we re-analyzed key experiments (**Fig. 3b**) with DIA-NN:

“To determine the extent to which these results were dependent on the software used, we re-analyzed the dilution series using DIA-NN. This yielded very similar results for input amounts of 1 μg with $\sim 4,000$ and $\sim 6,000$ Class 1 phosphosites identified by DIA-NN and Spectronaut, respectively, and $\sim 30\%$ difference in the number of localized phosphosites for 20 μg input (Appendix Fig. S2a). Comparing the base peptide sequences regardless of their modification, 84% of the peptides identified by DIA-NN overlapped with Spectronaut (Appendix Fig. S2b).”

Appendix Figure S2. Comparison of different processing software.

a Number of localized phosphosites (localization score >0.75) two alternative software packages (DIA-NN and Spectronaut (SN)) as a function of input amount of HeLa cell lysate.

b Overlap of base peptide sequences (i.e. regardless of modification and localization) identified by SN and DIA-NN from 20 μg input amount.

This is a larger overlap than reported by Mann and colleagues in the above paper using a library-based approach and appears more in line with a recent benchmarking study by Lou et al. (PMID: 36609502). While both programs are rapidly evolving, neither are open source, making it difficult to trace any discrepancy. However, considering the multiplicative gains of *μPhos*, this result shows that the performance of our workflow is consistent with different analysis software solutions.

DIA experiments are typically analyzed against experimental spectral libraries. Such project-specific libraries often result in high identification rates, but are difficult to transfer between laboratories. To eliminate experimental libraries as a potential confounder in our results, we chose to analyze all experiments using a ‘library-free’ approach (termed ‘directDIA’ in Spectronaut). Unfortunately, we found that on our computer, the processing times with DIA-NN in the library-free mode became surprisingly long when considering variable modifications (e.g. S/T/Y phosphorylation and M oxidation). This prevented us to re-analyze the entire dataset with DIA-NN in a reasonable timeframe. However, we have deposited all MS raw data in the PRIDE repository to facilitate re-analysis with alternative software solutions such as DIA-NN, OpenSWATH, MaxQuant, MS-Fragger, EncyclopeDIA or Skyline. The data are already fully accessible for reviewers, and we will release them to the community upon publication.

Likewise, information about the reagents used must be accurate. For instance, Empore is a product brand, not the name of the manufacturing and marketing company.

We thank the reviewer for spotting this. We have corrected the information in our revised manuscript and carefully checked all other items for accuracy.

Reviewer #3:

This is well-written manuscript by Oliinyk et al., reporting a method advance of phosphoproteomics for robust handling and analysis of samples with small quantity of proteins. The overall results presented are indeed quite impressive with >30,000 phosphopeptides quantified from as low as 20 ug starting materials. The utility of the workflow was further demonstrated in an application of time-resolved experiments of cells in response to tyrosine kinase inhibitors. Nevertheless, there are several limitations of this work, which prevents me to recommend for publication in its current format.

We thank the reviewer for their encouraging remarks on our manuscript. In the course of the revision, we performed additional experiments and analyses to address the remaining concerns as detailed below.

Main:

1. Despite the impressive results being results, the current work still appears to be a rather incremental advance over the authors' own previous work on *EasyPhos* platform (ref. 38). Conceptually, the current workflow is an integration of one-pot processing coupled with more advanced MS instrumentation. The improvement with one-pot processing (small volume) for phosphoproteomics has already been reported (PMID: 36653408).

We thank the reviewer for appreciating our results and regret that our original submission might have left the impression of otherwise rather small advances. Similar to the point raised by Reviewer #1, we acknowledge that our initial submission may not have sufficiently highlighted all advances and we have substantially revised our manuscript to address these points. In fact, our revised manuscript presents several valuable advances and novel aspects:

- Systematic optimization of experimental conditions to enrich phosphopeptides in low volumes with high selectivity.
- Comprehensive phosphoproteomics protocol for protein input amounts <20 μg in a rapid, robust and accessible plate format.
- Direct processing of cells and small tissue samples in parallel and in a single reaction vessel, i.e. without transfer or clean-up steps before the enrichment.
- Conceptually, these advances enable systematic perturbation experiments on a large scale, opening up for a wide range of applications in cell signaling research and, in a sense, challenging the standard choice of immunoblotting for studying cell signaling.
- Demonstration and validation of a 96-well perturbation experiment combined with phosphoproteomics.
- A new strategy to analyze variations in drug response profiles between drugs, demonstrating the possibility to gain biological insight from high-dimensional experimental designs.
- Mapping the phosphoproteome of small anatomical mouse brain regions.
- Discovery of heterogeneous kinase activity in subregions of the hippocampal formation in the mouse brain.

Specifically, to clarify the practical differences to *EasyPhos*, we summarize them just below:

Table. Comparison of *EasyPhos* and *μPhos*.

Feature	EasyPhos [Humphrey et al.]	μPhos [this study]
Phosphopeptide enrichment	TiO ₂ beads	TiO ₂ beads
Recommended input amount	200 μg	20 μg
Enrichment volume	800 μL	80 μL
Plate format	96 deep well	96-well flat-bottom, V-shape, U-shaped, deep well
Compatible lysis buffer	SDC	SDC, ACN, TFE

As discussed in our response to Reviewer #1, we think that the perceived ‘incremental advance’ over *EasyPhos* was exacerbated by us not clearly indicating or discussing our reference point. We clarified this in the revised version of our manuscript, for example:

“Investigating the raw intensities in more detail showed that the median phosphopeptide intensity increased 4-fold by decreasing the enrichment volume from 800 μL (i.e. EasyPhos) to 40 μL (Fig. EV2B).”

Comparing 800 μL (*EasyPhos*) enrichment volume vs. 80 μL (*μPhos*) on the same LC-MS platform resulted in:

- a 2-fold increase in the summed MS/MS intensity (**Fig. EV2a**)
- a 4-fold increase in phosphopeptide signal (**Fig. EV2b**)
- a 2-fold increase in the number of identified phosphopeptides from 20 μg HeLa lysate (**Fig. 2a**)
- a 2-3-fold increase in the number of identified phosphosites from 20 μg HeLa lysate (**Fig. EV2h**)

We are familiar with the work by Tsai et al. (PMID: 36653408) and have ensured that their contribution to the field is appropriately acknowledged in both the introduction and discussion sections of our revised manuscript. However, we disagree with the notion that only the first paper on a given topic can contain novelty. For example, there is now a large number of high-profile studies in the single-cell proteomics field that all work on the common theme of one-pot/small volume sample processing. We are of the opinion that the success of Tsai et al. and *μPhos* warrant further studies in this direction. That said, we emphasize that our implementation of the small-volume concept differs substantially from Tsai et al.:

- Tsai et al. propose a tip-based protocol, while *μPhos* is based on a plate format, which we believe has important implications for its scalability and accessibility (see also Reviewer #2).
- Both protocols perform cell lysis and digestion in small volumes, but only *μPhos* includes phosphopeptide enrichment in small volumes. In fact, optimizing these conditions turned out as a significant hurdle in developing the protocol.
- The point above was key to process samples directly in the very same reaction vessel, i.e. without a transfer step. In contrast, the Tsai et al. protocol involves an additional clean-up after the digestion, which is a source of potential sample losses.

2. It is unclear how much the improvement observed was resulting from the improved protocol or newer instrumentation. For example, in the authors' previous work, they reported the

quantification of ~10,000 phosphopeptides from ~25 μg proteins. It will be great if the authors can report the coverage using different instrumentation such as HFX vs TIMS-TOF, DIA vs. DDA. These data will be great for the broad readership. After all, the TIMS-TOF platform is still not widely available for most MS labs.

We thank the reviewer for pointing this out. Comparisons of phosphopeptide numbers between different protocols are indeed difficult, if possible at all, due to their dependence on many experimental parameters. This also applies to the mentioned 2018 study by Humphrey et al., which was based on a different cell line, different LC-MS setup, different data acquisition mode, and different analysis software. (Probably closest to our setup would be the 'match within replicates only' data point for the 60 min gradient and 25 μg starting material, resulting in ~7,000 phosphopeptides. In **Fig. 3b**, we now report >30,000 phosphosites (17,000 class I) from 20 μg starting material in the same analysis time.) Because of these considerations, we took care to disentangle the contributions of *μPhos* from all accompanying advances in MS instrumentation and data processing during the development of the protocol (**Fig. 2**). Please see also directly above as well as our response to Reviewer #1. All comparisons have been acquired on the very same MS platform, *i.e.* we expect them to be independent from the MS instrument. In particular, phosphoproteomics is typically limited by signal intensity, which correlates with spectral quality. For this reason, we are confident that the 4-fold increase in phosphopeptide signal (**Fig. EV2b**) transfers to other acquisition modes and instrument platforms. As discussed with the editor, we believe that this is the most appropriate approach to benchmark a new protocol, and it would in fact not be feasible for us to acquire data on a different instrument platform. We look forward to the community testing *μPhos* in different application areas and with different instrument setups (which some laboratories have already started based on our pre-print).

3. The data presented for the final application of time resolved phosphoproteomics were insufficient. As a reviewer, I was not able to examine the data quality from the presented information. I would suggest that the authors to make the following data available: a) the processed phosphor abundance values and associated statistics as supplementary data in spreadsheets; 2) heatmaps or volcano plots to illustrate the inhibitor responses as well as reproducibility across biological replicates.

Thank you. We apologize for the difficulties in examining the data in our original submission.

Following the reviewer's suggestion, we provide additional supplementary tables with the revised version of our manuscript. Because some of them are relatively large, we uploaded them to the PRIDE repository together with all associated raw files. If preferred otherwise, and space allows, we are happy to upload them in addition as Supplementary Data to the journal website.

Specifically, in response to this point we uploaded:

- *uPhos_CML_timeseries_ttest_tables.zip* (Tabular output of pairwise t-tests of the time-resolved tyrosine kinase inhibitor experiment.)

The data can be accessed via <https://www.ebi.ac.uk/pride/archive/projects/PXD043370>

Reviewer account (<https://www.ebi.ac.uk/pride/login>)

Username: reviewer_pxd043370@ebi.ac.uk

Password: IjvWPMeL

In response to the reviewer's second point, we performed the heatmap and volcano analysis as suggested (new **Appendix Figure S4**). Please note that the reproducibility across biological replicates is also shown in **Fig. 5c** (median Pearson correlation of 0.91).

Figure 5c. Pairwise Pearson correlation of biological replicates, across treatment time course and between drugs at each time point.

In general, we found that these additional analyses corroborated the findings from our initial principal component analysis:

“Consistent with this, the replicates grouped together in a principal component analysis and hierarchical clustering, with most variability attributed to the time course, while the comparison between proteome and phosphoproteome revealed a very weak correlation, which suggests that the observed changes in the phosphoproteome were not driven by protein abundance (Fig. EV4G, H, Appendix Fig. S4).”

To better illustrate the additional information gained from high-dimensional experiments compared to conventional measurements of log-fold changes at a single time-point and for a single drug, we developed an analysis strategy based on the time-weighted area under the curve as described in the new section **“Time-course analysis highlights drug-specific response signatures”**, new **Fig. 6**) in our revised manuscript.

Taken together, we believe that this application of μ Phos is a compelling use case to demonstrate the feasibility and potential to generate new biological insights by combining 96-well perturbation experiments with phosphoproteomics.

Appendix Figure S4. Statistical analysis of drug response signatures with *μPhos*.

a Pairwise volcano plots of all conditions (p adjusted < 0.05 , absolute fold change > 1.5)

b Heatmap of ANOVA-significant phosphosites across all conditions (FDR < 0.01).

Minor:

1. The color coding in Figure 6A is not apparent. May consider alternatives such as mini bar graphs.

We thank the reviewer for this suggestion. After discussing several options, we decided to simplify the color-coding in the figure (now **Fig. 5d**) but keep the circular pictograms to indicate the time component.

2. Error bars in Figure 6B might be helpful.

We added error bars to **Fig. 6b** of the revised manuscript.

3. It will be good to confirm the amounts of proteins for 50,000 cells in the final experiments.

Starting from 50,000 seed cells, we estimate ~70,000 cells at the time of sampling. This yielded ~20 μg of protein as estimated by a NanoDrop measurement from the flow-through of the enrichment.

30th Apr 2024

Manuscript Number: MSB-2023-11964RR

Title: μ Phos: a scalable and sensitive platform for high-dimensional phosphoproteomics

Dear Prof Meier,

Thank you for the submission of your revised manuscript to Molecular Systems Biology. I am pleased to inform you that we will be able to accept your manuscript pending the following final amendments and appropriate response to reviewers:

- 1) Please provide the "Author Checklist" and complete all relevant questions.
- 2) Please provide your manuscript as a .docx file (not PDF) without figures and track changes.
- 3) Data availability: Please ensure that the phosphoproteomics data in PRIDE are made freely accessible prior to publication.
- 4) Please rename "Competing Interests" to "Disclosure and competing interests statement". We updated our journal's competing interests policy in January 2022 and request authors to consider both actual and perceived competing interests. Please review the policy <https://www.embopress.org/competing-interests> and update your competing interests if necessary.
- 5) Author contributions: Please remove from the manuscript and specify author contributions in our submission system. CRediT has replaced the traditional author contributions section because it offers a systematic machine-readable author contributions format that allows for more effective research assessment. You are encouraged to use the free text boxes beneath each contributing author's name to add specific details on the author's contribution. More information is available in our guide to authors:

<https://www.embopress.org/page/journal/17574684/authorguide#authorshipguidelines>

- 6) Please correct the reference citation in the reference list. References should be alphabetical, not numerical. Where there are more than 10 authors on a paper, note that only 10 will be listed, followed by "et al.". Please check "Author Guidelines" for more information.

<https://www.embopress.org/page/journal/17574684/authorguide#referencesformat>

- 7) In the Materials and Methods, please take care of the following:

- Please re-structure your Materials and Methods as follows: after the Materials and Methods headline, there should be the sub-headline 'Reagents and Tools table' along with the table itself, followed by a section with the sub-headline 'Methods and Protocols', where all the written methods and protocols are included
 - Animals: Please ensure that an ethics statement and the approval committee for research on animals is included in the section where animal experiments are described in the Materials and Methods. Please also ensure that housing conditions as well as gender, age, and origin of the animals involved in experiments is reported.
 - Cell lines: Please include all information requested in the author checklist for cell lines used in the manuscript (accession number in repository or supplier name, catalog number, clone number, and/or RRID). Please also be sure to include a sentence in the Materials and Methods as to whether or not the cell lines were recently authenticated and tested for mycoplasma contamination.
 - Please ensure that a statement on whether or not blinding was done is included in the Materials and Methods even if no blinding was done.
 - Antibodies: please ensure that company name, catalog number, and dilutions/amounts of each antibody are reported.
- 8) Mass spectrometry: Please edit your manuscript to include the details of the mass spectrometry in the Methods as follows:
 - When describing the LC-MS experiments, please state the total number of samples analyzed, numbers and types of controls, number of technical and/or biological replicates (even if n=1)
 - Please describe all relevant parameters of the LC-MS experiment (particularly LC gradient, gas phase fragmentation settings, mass resolution of the MS1 and MS2 scans, etc)
 - Please provide a detailed description of the database search parameters and acceptance criteria used for peptide identification (name and version of identification software, name and version of protein sequence database, protease cleavage sites and number of missed cleavages, fixed and variable modifications, mass tolerance for precursor and fragment ions, minimum peptide length, any applied score cutoffs, peptide- and protein-level FDR, minimum number of unique peptides for protein identification)
 - Please describe and/or reference the statistical tests used for the proteomics data analyses.

- 9) Please place individual sections of the manuscript in the following order: Title page - Abstract & Keywords - Introduction - Results - Discussion - Materials and Methods - Data Availability - Acknowledgements - Disclosure and Competing Interests Statement - References - Figure Legends - Expanded View Figure Legends.

- 10) Please remove all figures from main manuscript file and leave only main and EV figure legends placed after the references. The main and EV figures should be uploaded as high-resolution individual figure files.

- 11) During a standard image analysis, we detected a possible re-use of image for the GAPDH controls in Figure 5C, and we would like to clarify these issues. Please check the composition of this figure. If you make any changes to the figure set, please describe what you have changed and why. If purposeful re-use of the image has occurred, please state this clearly in the figure legend. Please also insure that the source data for this figure is provided (see our request in Point 15).

- 12) For the figures and figure legends, please take care of the following:

- Please note that the legend for figure panel 5e is not labelled in the figure, however the corresponding legend for the same is labelled in the manuscript. This needs to be rectified.
- Please indicate the statistical test used for data analysis in the legend of figure EV 2f.
- Please note that the box plots need to be defined in terms of minima, maxima, centre, bounds of box and whiskers, and percentile in the legends of figures 3a; 5c; EV 2b-c, g, i; EV 3a; EV 5a.
- Please note that information related to n is missing in the legends of figures 2a; 3a; 5c; 6g; EV 2b-c, i; EV 3a; EV 5a.
- Please note that the error bars are not defined in the legends of figures 2a; 6g.

13) Appendix file: Please remove the EV figures and legends from the Appendix PDF. Please also ensure that the Table of Contents includes page numbers.

14) Synopsis:

- Synopsis image: Please upload a graphic that summarises the main findings of the manuscript on a glance. The synopsis image should be provided as a high-resolution jpeg file 550 pixels wide x (250-400) pixels high.
- Synopsis text: Please provide a short standfirst (maximum of 300 characters, including space), limit the bullet points to max. 5 and upload it as a separate .doc file. Please write the bullet points to summarise the key NEW findings. They should be designed to be complementary to the abstract - i.e. not repeat the same text. We encourage inclusion of key acronyms and quantitative information (maximum of 30 words / bullet point). Please use the passive voice.
- Please check your synopsis text and image before submission with your revised manuscript. Please be aware that in the proof stage minor corrections only are allowed (e.g., typos).

15) Source Data: Please ensure that a completed Source Data checklist is uploaded, along with a single source data file (zipped) per figure, with the panels clearly visible in the folder structure.

16) As part of the EMBO Publications transparent editorial process initiative (see our policy here: https://www.embopress.org/transparent-process#Review_Process), Molecular Systems Biology will publish online a Peer Review File (PRF) to accompany accepted manuscripts. This file will be published in conjunction with your paper and will include the anonymous referee reports, your point-by-point response and all pertinent correspondence relating to the manuscript. Let us know whether you agree with the publication of the PRF and as here, if you want to remove or not any figures from it prior to publication. Please note that the Authors checklist will be published at the end of the PRF.

17) Please provide a point-by-point letter INCLUDING my comments as well as the reviewer's reports and your detailed responses (as Word file).

I look forward to reading a new revised version of your manuscript as soon as possible.

Yours sincerely,

Poonam Bheda, PhD
Scientific Editor
Molecular Systems Biology

Please click on the link below to submit the revision.

Reviewer #1:

While the authors have included new experiments (e.g., analysis of some mouse tissues in Fig. 4, a new Western blot in Fig. 5, and data analysis in Fig. 6), our original major concerns regarding novelty and significance remain. We acknowledge improvements in the manuscript and the revision efforts; however, we still believe this paper would be better suited to a more specialized journal, rather than as a "Method" article in MSB, which demands significant novelty.

Furthermore, despite the precedence set by Tsai et al. (Ref 38), a very similar paper using 30 µg of peptides for phosphoproteomics was published in MCP (38548019). It should be noted that the bioRxiv date for µPhos predates this paper.

Other comments:

The manuscript highlights the 96-well plate format as a significant novelty. However, the practical significance of this approach for different types of samples is unclear. For instance, in the new mouse tissue experiments, tissues still require homogenization followed by high-energy sonication, processes that cannot be performed in 96-well plates. The structured protocol should include specific tips, such as the use of a multi-channel pipette during the transfer of buffers and bead suspensions-how is efficient pipetting ensured for bead suspensions?

The manuscript does not clearly disclose how many replicates were performed for each optimization step; it appears that some experiments, such as those shown in Figures S1d and S1g, were conducted only once (N=1)? The authors should provide more

transparency regarding the replicate details of protocol optimization.

The analysis employing pAUC to identify drug-specific response signatures fails to account for the longitudinal process and time course. Different phosphosites regulated at various times may yield identical pAUC values. The methods used for bioinformatic annotation of pathways are also unclear-are they performed at the site level or protein level? Additionally, the processing of phosphosite-level data for biological quantification is ambiguous and needs further clarification. It seems the localization score was activated and kept as 0??

Reviewer #2:

This paper is the revised version of MSB-2023-11964, and authors significantly modified the original version especially for the applicability to the tissue samples. Although this reviewer had already given a positive evaluation of the previous version, he believes that this revision has expanded the applicability of μ Phos and further empowered its significance as a method. The problems pointed out in the previous version were also appropriately addressed by adding results from DIA-NN.

Reviewer #3:

I really appreciate the efforts of the authors in addressing the reviewers' concerns and in revising the manuscript into a much improved product. I have no reservation in recommending it for publication.

I only have a minor point for clarification:

1. In Figure 5a, it was highlighted that 196 samples need only 96h LC-MS time for phosphoproteome samples. Could the authors clarify in their LC-MS method section about the LC gradient length and total cycle time per run?

Manuscript MSB-2023-11964, *μPhos*: a scalable and sensitive platform for high-dimensional phosphoproteomics

Point-by-point response

We thank the editor and reviewers for their detailed feedback on our revised manuscript. We are delighted by the positive assessment of Reviewers #2 and #3 and hope the final revisions detailed below address the remaining points of Reviewer #1 to their satisfaction.

Editor:

- 1) Please provide the "Author Checklist" and complete all relevant questions.

Done.

- 2) Please provide your manuscript as a .docx file (not PDF) without figures and track changes.

Done.

- 3) Data availability: Please ensure that the phosphoproteomics data in PRIDE are made freely accessible prior to publication.

Done.

- 4) Please rename "Competing Interests" to "Disclosure and competing interests statement". We updated our journal's competing interests policy in January 2022 and request authors to consider both actual and perceived competing interests. Please review the policy <https://www.embopress.org/competing-interests> and update your competing interests if necessary.

Done.

- 5) Author contributions: Please remove from the manuscript and specify author contributions in our submission system. CRediT has replaced the traditional author contributions section because it offers a systematic machine-readable author contributions format that allows for more effective research assessment. You are encouraged to use the free text boxes beneath each contributing author's name to add specific details on the author's contribution. More information is available in our guide to authors:

<https://www.embopress.org/page/journal/17574684/authorguide#authorshipguidelines>

Done.

- 6) Please correct the reference citation in the reference list. References should be alphabetical, not numerical. Where there are more than 10 authors on a paper, note that only 10 will be listed, followed by "et al.". Please check "Author Guidelines" for more information.

<https://www.embopress.org/page/journal/17574684/authorguide#referencesformat>

Done.

- 7) In the Materials and Methods, please take care of the following:

- Please re-structure your Materials and Methods as follows: after the Materials and Methods headline, there should be the sub-headline 'Reagents and Tools table' along with the table

itself, followed by a section with the sub-headline 'Methods and Protocols', where all the written methods and protocols are included

Done.

- Animals: Please ensure that an ethics statement and the approval committee for research on animals is included in the section where animal experiments are described in the Materials and Methods. Please also ensure that housing conditions as well as gender, age, and origin of the animals involved in experiments is reported.

Done.

- Cell lines: Please include all information requested in the author checklist for cell lines used in the manuscript (accession number in repository or supplier name, catalog number, clone number, and/or RRID). Please also be sure to include a sentence in the Materials and Methods as to whether or not the cell lines were recently authenticated and tested for mycoplasma contamination.

Done.

- Please ensure that a statement on whether or not blinding was done is included in the Materials and Methods even if no blinding was done.

Done.

- Antibodies: please ensure that company name, catalog number, and dilutions/amounts of each antibody are reported.

Done.

8) Mass spectrometry: Please edit your manuscript to include the details of the mass spectrometry in the Methods as follows:

- When describing the LC-MS experiments, please state the total number of samples analyzed, numbers and types of controls, number of technical and/or biological replicates (even if n=1)

We added this information to the corresponding figure legends.

- Please describe all relevant parameters of the LC-MS experiment (particularly LC gradient, gas phase fragmentation settings, mass resolution of the MS1 and MS2 scans, etc)

Done (sub-section Liquid chromatography-mass spectrometry).

- Please provide a detailed description of the database search parameters and acceptance criteria used for peptide identification (name and version of identification software, name and version of protein sequence database, protease cleavage sites and number of missed cleavages, fixed and variable modifications, mass tolerance for precursor and fragment ions, minimum peptide length, any applied score cutoffs, peptide- and protein-level FDR, minimum number of unique peptides for protein identification)

Done (sub-section Raw data processing).

- Please describe and/or reference the statistical tests used for the proteomics data analyses.

Done (sub-section Bioinformatics).

- 9) Please place individual sections of the manuscript in the following order: Title page - Abstract & Keywords - Introduction - Results - Discussion - Materials and Methods - Data Availability - Acknowledgements - Disclosure and Competing Interests Statement - References - Figure Legends - Expanded View Figure Legends.

Done.

- 10) Please remove all figures from main manuscript file and leave only main and EV figure legends placed after the references. The main and EV figures should be uploaded as high-resolution individual figure files.

Done.

- 11) During a standard image analysis, we detected a possible re-use of image for the GAPDH controls in Figure 5C, and we would like to clarify these issues. Please check the composition of this figure. If you make any changes to the figure set, please describe what you have changed and why. If purposeful re-use of the image has occurred, please state this clearly in the figure legend. Please also insure that the source data for this figure is provided (see our request in Point 15).

Thank you for spotting this. For every Western Blot, we ran a GAPDH control as a loading control and for internal normalization. If applicable, membranes were cut and stripped to detect multiple proteins and phosphosites in one run. However, when we composed the original figure panel, the individual protein/phosphosite bands were rearranged and a representative image of the GAPDH control was inadvertently re-used. To avoid any confusion, we have reassembled the panel and clarify in the figure legend that representative blots are shown. We provide the underlying source data for all replicates.

- 12) For the figures and figure legends, please take care of the following:

- Please note that the legend for figure panel 5e is not labelled in the figure, however the corresponding legend for the same is labelled in the manuscript. This needs to be rectified.

Done.

- Please indicate the statistical test used for data analysis in the legend of figure EV 2f.

Done.

- Please note that the box plots need to be defined in terms of minima, maxima, centre, bounds of box and whiskers, and percentile in the legends of figures 3a; 5c; EV 2b-c, g, i; EV 3a; EV 5a.

Done.

- Please note that information related to n is missing in the legends of figures 2a; 3a; 5c; 6g; EV 2b-c, i; EV 3a; EV 5a.

Added.

- Please note that the error bars are not defined in the legends of figures 2a; 6g.

Added.

- 13) Appendix file: Please remove the EV figures and legends from the Appendix PDF. Please also ensure that the Table of Contents includes page numbers.

Done.

- 14) Synopsis:

- Synopsis image: Please upload a graphic that summarises the main findings of the manuscript on a glance. The synopsis image should be provided as a high-resolution jpeg file 550 pixels wide x (250-400) pixels high.

- Synopsis text: Please provide a short standfirst (maximum of 300 characters, including space), limit the bullet points to max. 5 and upload it as a separate .doc file. Please write the bullet points to summarise the key NEW findings. They should be designed to be complementary to the abstract - i.e. not repeat the same text. We encourage inclusion of key acronyms and quantitative information (maximum of 30 words / bullet point). Please use the passive voice.

Done.

- 15) Source Data: Please ensure that a completed Source Data checklist is uploaded, along with a single source data file (zipped) per figure, with the panels clearly visible in the folder structure.

Done.

- 16) As part of the EMBO Publications transparent editorial process initiative (see our policy here: https://www.embopress.org/transparent-process#Review_Process), Molecular Systems Biology will publish online a Peer Review File (PRF) to accompany accepted manuscripts. This file will be published in conjunction with your paper and will include the anonymous referee reports, your point-by-point response and all pertinent correspondence relating to the manuscript. Let us know whether you agree with the publication of the PRF and as here, if you want to remove or not any figures from it prior to publication. Please note that the Authors checklist will be published at the end of the PRF.

We agree with the publication of the PRF. We do not ask to remove any figures from it prior to publication.

- 17) Please provide a point-by-point letter INCLUDING my comments as well as the reviewer's reports and your detailed responses (as Word file).

Done.

Reviewer #1:

While the authors have included new experiments (e.g., analysis of some mouse tissues in Fig. 4, a new Western blot in Fig. 5, and data analysis in Fig. 6), our original major concerns regarding novelty and significance remain. We acknowledge improvements in the manuscript and the revision efforts; however, we still believe this paper would be better suited to a more specialized journal, rather than as a "Method" article in MSB, which demands significant novelty.

We thank the reviewer for acknowledging our efforts to address their concerns during the revision process. We believe that the additional experiments and analyses in response to this reviewer's previous points have significantly strengthened our manuscript and helped us to highlight novel aspects as well as its potential impact on systems biology. We have done our best to clarify the remaining points below.

Furthermore, despite the precedence set by Tsai et al. (Ref 38), a very similar paper using 30 μg of peptides for phosphoproteomics was published in MCP (38548019). It should be noted that the bioRxiv date for μPhos predates this paper.

Please note that the paper by Bortel et al. was published after we submitted our revised manuscript. We added the reference to this version of our manuscript.

Other comments:

The manuscript highlights the 96-well plate format as a significant novelty. However, the practical significance of this approach for different types of samples is unclear. For instance, in the new mouse tissue experiments, tissues still require homogenization followed by high-energy sonication, processes that cannot be performed in 96-well plates. The structured protocol should include specific tips, such as the use of a multi-channel pipette during the transfer of buffers and bead suspensions-how is efficient pipetting ensured for bead suspensions?

Thank you for raising this point. We would like to clarify that we performed two distinct mouse tissue experiments with slightly different objectives:

- 1) In response to this Reviewer, we first assessed the digestion and enrichment efficiency for complex samples (Fig. EV2H-J). To eliminate tissue lysis as a potential confounding factor in this experiment, we started with bulk tissue lysates prepared using a standard protocol. (Methods - Tissue lysis for bulk experiments)
- 2) Further, to demonstrate the applicability of the full protocol to tissue samples, we analyzed small anatomical regions of the mouse brain in 96-well format (Fig. 4). This experiment did not require tissue homogenization and high-energy sonication (Fig. EV3 A,B). (Methods - Dissection and lysis of mouse brain tissue)

In our experience minimal starting amounts facilitate protein extraction, however, we acknowledge that other sample types may still result in incomplete lysis under these conditions and had already noted this as a potential limitation in the Discussion. To specify this point, we revised this paragraph and added references for 96-well ultrasonication and homogenization:

"Under these conditions, we also found that the duration of tryptic digestion could be reduced from overnight to just 4 hours for cell culture and fresh-frozen brain tissue samples. We note that other sample types (e.g. muscle biopsies or formalin-fixed paraffin embedded tissue) and higher input amounts might still benefit from longer digestion or require harsher conditions for efficient lysis and protein extraction. In this case, adapting the digestion time

as needed and complementing the protocol with ultra-sonication or mechanical cell lysis in plate format, e.g. similar to (Müller et al, 2020; Michaelis et al, 2023), should be straightforward, as we found that the low working volumes are compatible with different lysis buffers and 96-well plates.”

Thank you for the suggestion to include tips in the structured protocol. We revised this section accordingly and added a new Supplementary Figure to illustrate recommended aspiration points for different well formats.

The manuscript does not clearly disclose how many replicates were performed for each optimization step; it appears that some experiments, such as those shown in Figures S1d and S1g, were conducted only once (N=1)? The authors should provide more transparency regarding the replicate details of protocol optimization.

Thank you. In accordance with the journal's guidelines, the number of replicates is now indicated in each figure legend.

The analysis employing pAUC to identify drug-specific response signatures fails to account for the longitudinal process and time course. Different phosphosites regulated at various times may yield identical pAUC values.

The reviewer is correct that, only considering the area under the curve, two phosphosites regulated at different times could potentially result in the same value. For this reason, and to prioritize fast-responding sites, we calculated a time-weighted area under the curve by multiplying the area with a factor T_{max}/T ('pAUC', formula in Fig. 6A). We found that this intuitively ranks key phosphosites and provides biologically relevant insight into the cellular response to tyrosine kinase inhibitors, while providing a straightforward means of comparing different drugs (as initially requested by this reviewer). However, as there are certainly more questions that could be explored from this dataset, we hope that making all the data accessible will facilitate re-analysis by the community.

The methods used for bioinformatic annotation of pathways are also unclear-are they performed at the site level or protein level? Additionally, the processing of phosphosite-level data for biological quantification is ambiguous and needs further clarification. It seems the localization score was activated and kept as 0??

Thank you. In response to this point we extended the 'Bioinformatics' section in 'Methods and Protocols' and added further details to the 'Raw data processing' section.

Reviewer #2:

This paper is the revised version of MSB-2023-11964, and authors significantly modified the original version especially for the applicability to the tissue samples. Although this reviewer had already given a positive evaluation of the previous version, he believes that this revision has expanded the applicability of *μPhos* and further empowered its significance as a method. The problems pointed out in the previous version were also appropriately addressed by adding results from DIA-NN.

Thank you for your critical insight and support of our manuscript.

Reviewer #3:

I really appreciate the efforts of the authors in addressing the reviewers' concerns and in revising the manuscript into a much improved product. I have no reservation in recommending it for publication.

Thank you. We highly appreciate the constructive feedback in the review process.

I only have a minor point for clarification:

1. In Figure 5a, it was highlighted that 196 samples need only 96h LC-MS time for phosphoproteome samples. Could the authors clarify in their LC-MS method section about the LC gradient length and total cycle time per run?

We have clarified in the figure that we refer to 96 proteome and 96 phosphoproteome measurements and added the LC gradient information to the 'Methods and Protocols' section.

4th Jun 2024

Manuscript Number: MSB-2023-11964RRR

Title: μ Phos: a scalable and sensitive platform for high-dimensional phosphoproteomics

Dear Prof Meier,

Thank you again for submitting your work to Molecular Systems Biology.

Before proceeding with your manuscript, we note that there are still a few points that need to be resolved. Please address the following, including a response to each point in your resubmission:

- There appears to be a missing reference in the legend for Figure EV2 panel F ("Kolmogorov-Smirnov test [ref]").
- Thank you for your explanation of the re-use of the GAPDH loading control in Figure 5E. However from your explanation, the figure legends, and the Source Data files that you provided, it is unclear whether the images shown for the different proteins in Figure 5E come from the same or from different blots. If they come from different blots, showing a single GAPDH loading control as representative is not sufficient. Each panel that comes from a different blot is required to have its own loading control. Please edit the figure and the manuscript. We would also advise labelling the Source Data files in a way that make it easier to understand which panels are from the same blots (i.e. so that the reader can identify which protein of interest image corresponds to which GAPDH control).
- The size and dimensions of the Synopsis Image do not fit to our requirements. Please provide a synopsis image a high-resolution jpeg file 550 pixels wide x (250-400) pixels high. We would be able to resize the figure ourselves, but due to the square shape of the figure, when resized to 550 pixels wide, the figure is too high.

Click on the link below to submit your revised paper.

Thank you for submitting this paper to Molecular Systems Biology. If you have any questions about the requests above, please don't hesitate to ask.

Yours sincerely,

Poonam Bheda, PhD
Scientific Editor
Molecular Systems Biology

Point-by-point response

Editor:

- 1) There appears to be a missing reference in the legend for Figure EV2 panel F ("Kolmogorov-Smirnov test [ref]").

Thank you. As the Kolmogorov-Smirnov test is a common statistical test described in textbooks and easily identified by name, we have decided to omit this reference from the figure legend.

- 2) Thank you for your explanation of the re-use of the GAPDH loading control in Figure 5E. However from your explanation, the figure legends, and the Source Data files that you provided, it is unclear whether the images shown for the different proteins in Figure 5E come from the same or from different blots. If they come from different blots, showing a single GAPDH loading control as representative is not sufficient. Each panel that comes from a different blot is required to have its own loading control. Please edit the figure and the manuscript. We would also advise labelling the Source Data files in a way that make it easier to understand which panels are from the same blots (i.e. so that the reader can identify which protein of interest image corresponds to which GAPDH control).

Thank you for the clarification. We edited the figure in the newly revised version accordingly. The revised figure shows images of three different blots grouped together with the corresponding Gapdh loading controls. (Please note that when grouping them together we replaced the image for Stat5 with another replicate.)

We edited the figure legend accordingly:

e Validation of selected phosphorylation sites in downstream BCR::ABL1 signaling by immunoblotting. The figure shows representative blots from n=3 replicates and the corresponding Gapdh loading controls. The line plots show the median fold-change relative to Gapdh. Whiskers indicate the absolute inter-replicate variance (n=4 for MS, n=3 for immunoblotting).

We added the following sentence to the Methods section:

For each blot, we measured Gapdh as a loading control and for internal normalization. If applicable, membranes were cut and stripped to detect multiple proteins and phosphosites in one run.

To make our Source Data easier to understand we added a new Excel sheet indicating replicates, file names and corresponding loading controls.

- 3) The size and dimensions of the Synopsis Image do not fit to our requirements. Please provide a synopsis image a high-resolution jpeg file 550 pixels wide x (250-400) pixels high. We would be able to resize the figure ourselves, but due to the square shape of the figure, when resized to 550 pixels wide, the figure is too high.

We uploaded a new Synopsis Image with the correct size and dimensions.

11th Jun 2024

Manuscript number: MSB-2023-11964RRRR

Title: μ Phos: a scalable and sensitive platform for high-dimensional phosphoproteomics

Dear Prof Meier,

Thank you again for sending us your revised manuscript. We are now satisfied with the modifications made and I am pleased to inform you that your paper has been accepted for publication.

Yours sincerely,

Poonam Bheda, PhD
Scientific Editor
Molecular Systems Biology
